# PROVABLY ACCURATE SHAPLEY VALUE ESTIMATION VIA LEVERAGE SCORE SAMPLING

**Christopher Musco**
New York University
cmusco@nyu.edu

**R. Teal Witter**
New York University
rtealwitter@nyu.edu

## ABSTRACT

Originally introduced in game theory, Shapley values have emerged as a central tool in explainable machine learning, where they are used to attribute model predictions to specific input features. However, computing Shapley values exactly is expensive: for a general model with $n$ features, $O(2^n)$ model evaluations are necessary. To address this issue, approximation algorithms are widely used. One of the most popular is the Kernel SHAP algorithm, which is model agnostic and remarkably effective in practice. However, to the best of our knowledge, Kernel SHAP has no strong non-asymptotic complexity guarantees. We address this issue by introducing *Leverage SHAP*, a lightweight modification of Kernel SHAP that provides provably accurate Shapley value estimates with just $O(n \log n)$ model evaluations. Our approach takes advantage of a connection between Shapley value estimation and agnostic active learning by employing *leverage score sampling*, a powerful regression tool. Beyond theoretical guarantees, we find that Leverage SHAP achieves an approximately $50\%$ reduction in error compared to the highly optimized implementation of Kernel SHAP in the widely used SHAP library [Lundberg & Lee, 2017].

## 1 INTRODUCTION

While AI is increasingly deployed in high-stakes domains like education, healthcare, finance, and law, increasingly complicated models often make predictions or decisions in an opaque and uninterpretable way. In high-stakes domains, transparency in a model is crucial for building trust. Moreover, for researchers and developers, understanding model behavior is important for identifying areas of improvement and applying appropriate safeguards. To address these challenges, Shapley values have emerged as a powerful game-theoretic approach for interpreting even opaque models (Shapley, 1951; Štrumbelj & Kononenko, 2014; Datta et al., 2016; Lundberg & Lee, 2017). These values can be used to effectively quantify the contribution of each input feature to a model's output, offering at least a partial, principled explanation for why a model made a certain prediction.

Concretely, Shapley values originate from game-theory as a method for determining fair 'payouts' for a cooperative game involving $n$ players. The goal is to assign higher payouts to players who contributed more to the cooperative effort. Shapley values quantify the contribution of a player by measuring how its addition to a set of other players changes the value of the game. Formally, let the *value function* $v : 2^{[n]} \to \mathbb{R}$ be a function defined on sets $S \subseteq [n]$. The Shapley value for player $i$ is:

$$\phi_i = \frac{1}{n} \sum_{S \subseteq [n] \setminus \{i\}} \frac{v(S \cup \{i\}) - v(S)}{\binom{n-1}{|S|}}. \tag{1}$$

The denominator weights the marginal contribution of player $i$ to set $S$ by the number of sets of size $|S|$, so that the marginal contribution to sets of each size are equally considered. With this weighting, Shapley values are known to be the unique values that satisfy four desirable game-theoretic properties: Null Player, Symmetry, Additivity, and Efficiency (Shapley, 1951). For further details on Shapley values and their theoretical motivation, we refer the reader to Molnar (2024).

A popular way of using Shapley values for explainable AI is to attribute predictions made by a model $f : \mathbb{R}^n \to \mathbb{R}$ on a given input $\mathbf{x} \in \mathbb{R}^n$ compared to a baseline input $\mathbf{y} \in \mathbb{R}^n$ (Lundberg & Lee,

2017). The players are the features and $v(S)$ is the prediction of the model when using the features in $S$; i.e., $v(S) = f(\mathbf{x}^S)$ where $\mathbf{x}_i^S = x_i$ for $i \in S$ and $\mathbf{x}_i^S = y_i$ otherwise.[1] When $v$ is defined in this way, Shapley values measure how each feature value in the input contributes to the prediction.

Shapley values also find other applications in machine learning, including in feature and data selection. For feature selection, the value function $v(S)$ is taken to be the model loss when when using the features in $S$ (Marcílio & Eler, 2020; Fryer et al., 2021). For data selection, $v(S)$ is taken to be the loss when using the data observations in $S$ (Jia et al., 2019; Ghorbani & Zou, 2019).

## 1.1 EFFICIENT SHAPLEY VALUE COMPUTATION

Naively, computing all $n$ Shapley values according to Equation 1 requires $O(2^n)$ evaluations of $v$ (each of which involves the evaluation of a learned model) and $O(2^n)$ time. This cost can be reduced in certain special cases, e.g. when computing feature attributions for linear models or decision trees (Lundberg et al., 2018; Campbell et al., 2022; Amoukou et al., 2022; Chen et al., 2018).

More often, when $v$ is based on an arbitrary model, like a deep neural network, the exponential cost in $n$ is avoided by turning to approximation algorithms for estimating Shapley values, including sampling, permutation sampling, and Kernel SHAP (Strumbelj & Kononenko, 2010; Lundberg & Lee, 2017; Mitchell et al., 2022). The Kernel SHAP method is especially popular, as it performs well in practice for a variety of models, requiring just a small number of black-box evaluations of $v$ to obtain accurate estimates to $\phi_1, \ldots, \phi_n$. The method is a corner-stone of the ubiquitous SHAP library for explainable AI based on Shapley values (Lundberg & Lee, 2017).

Kernel SHAP is based on an elegant connection between Shapley values and least squares regression (Charnes et al., 1988). Specifically, let $[n]$ denote $\{1, \ldots, n\}$, $\emptyset$ denote the empty set, and $\mathbf{1}$ denote an all 1's vector of length $n$. The Shapley values $\boldsymbol{\phi} = [\phi_1, \ldots, \phi_n] \in \mathbb{R}^n$ are known to satisfy:

$$\boldsymbol{\phi} = \underset{\mathbf{x}:\langle \mathbf{x}, \mathbf{1}\rangle = v([n]) - v(\emptyset)}{\arg\min} \|\mathbf{Z}\mathbf{x} - \mathbf{y}\|_2, \tag{2}$$

where $\mathbf{Z} \in \mathbb{R}^{2^n - 2 \times n}$ is a specific structured matrix whose rows correspond to sets $S \subseteq [n]$ with $0 < |S| < n$, and $\mathbf{y} \in \mathbb{R}^{2^n - 2}$ is vector whose entries correspond to values of $v(S)$. (We precisely define $\mathbf{Z}$ and $\mathbf{y}$ in Section 2.)

Since solving the regression problem in Equation 2 directly would require evaluating $v(S)$ for all $2^n - 2$ subsets represented in $\mathbf{y}$, Kernel SHAP solves the problem approximately via *subsampling*. Concretely, for a given number of samples $m$ and a discrete probability distribution $\mathbf{p} \in [0, 1]^{2^n - 2}$ over rows in $\mathbf{Z}$, consider a sampling matrix $\mathbf{S} \in \mathbb{R}^{m \times 2^n - 2}$, where each row of $\mathbf{S}$ is 0 except for a single entry $1/\sqrt{p_j}$ in the $j^{\text{th}}$ entry with probability $p_j$. The Kernel SHAP estimate is given by

$$\tilde{\boldsymbol{\phi}} = \underset{\mathbf{x}:\langle \mathbf{x}, \mathbf{1}\rangle = v([n]) - v(\emptyset)}{\arg\min} \|\mathbf{S}\mathbf{Z}\mathbf{x} - \mathbf{S}\mathbf{y}\|_2. \tag{3}$$

Importantly, computing this estimate only requires at most $m$ evaluations of the value function $v$, since $\mathbf{S}\mathbf{y}$ can be constructed from observing at most $m$ entries in $\mathbf{y}$. Value function evaluations typically dominate the computational cost of actually solving the regression problem in Equation 3, so ideally $m$ is chosen as small as possible. In an effort to reduce sample complexity, Kernel SHAP does not sample rows uniformly. Instead, the row corresponding to subset $S$ is chosen with probability proportional to:

$$w(|S|) = \left( \binom{n}{|S|} |S|(n - |S|) \right)^{-1}. \tag{4}$$

The specific motivation for this distribution is discussed further in Section 2, but the choice is intuitive: the method is more likely to sample rows corresponding to subsets whose size is close to 0 or $n$, which aligns with the fact that these subsets more heavily impact the Shapley values (Equation 1). Ultimately, however, the choice of $w(|S|)$ is heuristic.

---

[1] There are multiple ways to define the baseline $\mathbf{y}$. The simplest is to consider a fixed vector (Lundberg & Lee, 2017). Other approaches define $\mathbf{y}$ as a random vector that is drawn from a data distribution (Lundberg et al., 2018; Janzing et al., 2020) and take $v(S) = \mathbb{E}[\mathbf{x}^S]$, where the expectation is taken over the random choice of $\mathbf{y}$. The focus of our work is estimating Shapley values once the values function $v$ is fixed, so our methods and theoretical analysis are agnostic to the specific approach used.

In practice, the Kernel SHAP algorithm is implemented with further optimizations (Lundberg & Lee, 2017; Covert & Lee, 2021; Jethani et al., 2021).

- **Paired Sampling:** If a row corresponding to set $S$ is sampled, the row corresponding to its complement $\bar{S} := [n] \setminus S$ is also sampled. This *paired* sampling strategy intuitively balances samples so each player $i$ is involved in the exact same number of samples. Paired sampling substantially improves Kernel SHAP (Covert & Lee, 2021).
- **Sampling without Replacement:** When $n$ is small, sampling with replacement can lead to inaccurate solutions even as the number of samples $m$ approaches and surpasses $2^n$, even though $2^n$ evaluations of $v$ are sufficient to exactly recover the Shapley values. In the SHAP library implementation of Kernel SHAP, this issue is addressed with a version of sampling without replacement: if there is a sufficient 'sample budget' for a given set size, all sets of that size are sampled (Lundberg & Lee, 2017). Doing so substantially improves Kernel SHAP as $m$ approaches $2^n$.

## 1.2 OUR CONTRIBUTIONS

Despite its ubiquity in practice, to the best of our knowledge, no non-asymptotic theoretical accuracy guarantees are known for Kernel SHAP when implemented with $m < 2^n - 2$ row samples (which corresponds to $m$ evaluations of the value function, $v$). We address this issue by proposing **Leverage SHAP**, a lightweight modification of Kernel SHAP that 1) enjoys strong theoretical accuracy guarantees and 2) consistently outperforms Kernel SHAP in experiments, achieving a $50\%$ reduction in error on average (see Table 2).

Leverage SHAP begins with the observation that a nearly optimal solution to the regression problem in Equation 2 can be obtained by sampling just $\tilde{O}(n)$ rows with probability proportional to their *statistical leverage scores*, a natural measure for the "importance" or "uniqueness" of a matrix row (Sarlós, 2006; Rauhut & Ward, 2012; Hampton & Doostan, 2015; Cohen & Migliorati, 2017).[2] This fact immediately implies that, *in principle*, we should be able to provably approximate $\phi$ with a nearly-linear number of value function evaluations (one for each sampled row).

However, leverage scores are expensive to compute, naively requiring at least $O(2^n)$ time to write down for a matrix like $\mathbf{Z}$ with $O(2^n)$ rows. Our key observation is that this bottleneck can be avoided in the case of Shapley value estimation: we prove that the leverage scores of $\mathbf{Z}$ have a simple closed form that admits efficient sampling without ever writing them all down. Concretely, we show that the leverage score of the row corresponding to any subset $S \subset [n]$ is proportional to $\binom{n}{|S|}^{-1}$.

This suggests a similar, but meaningfully different sampling distribution than the one used by Kernel SHAP (see Equation 4). Since all subsets of a given size have the same leverage score, we can efficiently sample proportional to the leverage scores by sampling a random size $s$ uniformly from $\{1, \ldots, n-1\}$ then selecting a subset $S$ of size $s$ uniformly at random.

**Theorem 1.1.** *For any $\epsilon > 0$ and constant $\delta > 0$, the Leverage SHAP algorithm uses $m = O(n \log(\frac{n}{\delta}) + n\frac{1}{\epsilon\delta})$ evaluations of $v$ in expectation and $O(mn^2)$ additional runtime to return estimated Shapley values $\tilde{\phi}$ satisfying $\langle \tilde{\phi}, \mathbf{1} \rangle = v([n]) - v(\emptyset)$ and, with probability $1 - \delta$,*

$$\|\mathbf{Z}\tilde{\phi} - \mathbf{y}\|_2^2 \leq (1 + \epsilon)\|\mathbf{Z}\phi - \mathbf{y}\|_2^2. \tag{5}$$

In words, Theorem 1.1 establishes that, with a near-linear number of function evaluations, we can compute approximate Shapley values whose *objective value* is close to the true Shapley values. We also require $O(mn^2)$ additional runtime to solve the linearly constrained regression problem in Equation 3, although this cost can be reduced in practice using, e.g., iterative methods.

By leveraging the fact that $\mathbf{Z}$ is a well-conditioned matrix, the bound in Equation 5 also implies a bound on the average squared error, $\|\tilde{\phi} - \phi\|_2^2$, which is provided in Section 4. (In Appendix C, we discuss why estimating Shapley values with multiplicative error is NP-hard.)

Beyond our theoretical results, we also show that leverage score sampling can be naturally combined with paired sampling, without sacrificing theoretical guarantees. Moreover, we show that

---

[2]Technically, this fact is known for *unconstrained* least square regression. Some additional work is needed to handle the linear constraint in Equation 2, but doing so is relatively straightforward.

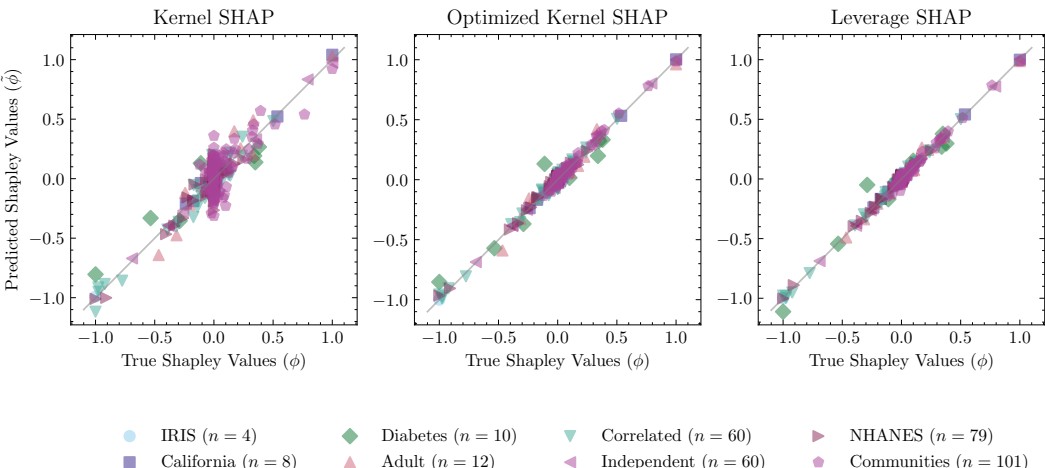

Figure 1: Predicted versus true Shapley values for all features in 8 datasets (we use $m = 5n$ samples for this experiment). Points near the identity line indicate that the estimated Shapley value is close to its true value. The plots suggest that our Leverage SHAP method is more accurate than the baseline Kernel SHAP algorithm, as well as the optimized Kernel SHAP implementation available in the SHAP library. We corroborate these findings with more experiments in Section 5.

a natural "without replacement" version of leverage score sampling leads to additional accuracy improvements when $m$ is large. Overall, these simple optimizations lead to an algorithm that consistently outperforms the optimized version of Kernel SHAP available in the SHAP Library (this 'Optimized Kernel SHAP' algorithm also uses paired sampling and sampling without replacement). We illustrate this point in Figure 1.

More extensive experiments are included in Section 5. We find that the improvement of Leverage SHAP over Kernel SHAP is especially substantial in settings where $n$ is large in comparison to $m$ and when we only have access to noisy estimates of the Shapley values. This is often the case in applications to explainable AI where, as discussion earlier, $v(S)$ is an expectation over functions involving random baselines and estimated via a finite sample.

## 1.3 RELATED WORK

**Shapley Values Estimation.** As discussed, naively computing Shapley values requires an exponential number of evaluations of $v$. While more efficient methods exist for certain structured value functions (see references in Section 1.1), for generic functions, faster algorithms involve some sort of approximation. The most direct way of obtaining an approximation to is approximate the summation definition of Shapley values (Equation 1), which involves $O(2^n)$ terms for each subset of $[n]$, with a random subsample of subsets (Castro et al., 2009; Strumbelj & Kononenko, 2010; Štrumbelj & Kononenko, 2014). The first methods for doing so use a different subsample for each player $i$.

A natural alternative is to try selecting subsets in such a way that allows them to be reused across multiple players (Illés & Kerényi, 2019; Mitchell et al., 2022). However, since each term in the summation involves both $v(S)$ and $v(S \cup \{i\})$, it is difficult to achieve high levels of sample reuse when working with the summation definition. One option is to split the sum into two, and separately estimate $\sum_{S \subseteq [n] \setminus \{i\}} v(S \cup \{i\}) / \binom{n-1}{|S|}$ and $\sum_{S \subseteq [n] \setminus \{i\}} v(S) / \binom{n-1}{|S|}$. This allows substantial subset reuse, but tends to perform poorly in practice due to high variance in the individual sums (Wang & Jia, 2023). By solving a global regression problem that determines the entire $\phi$ vector, Kernel SHAP can be viewed as a more effective way of reusing subsets to obtain Shapley values for all players.

In addition to the sampling methods referenced above, Covert & Lee (2021) propose a modified Kernel SHAP algorithm, prove it is unbiased, and compute its asymptotic variance. However, they find that the modified version performs worse than Kernel SHAP. Additionally, we note that some recent work in the explainable AI setting takes advantage of the fact that we often wish to evaluate feature impact for a large number of different input vectors $\mathbf{x} \in \mathbb{R}^n$, which each induce their own

set function $v$. In this setting, it is possible to leverage information gathered for one input vector to more efficiently compute Shapley values for another, further reducing sample cost (Schwab & Karlen, 2019; Jethani et al., 2021). This setting is incomparable to ours; like Kernel SHAP, Leverage SHAP works in the setting where we have black-box access to a *single* value function $v$.

**Leverage Scores.** As discussed, leverage scores are a natural measure of importance for matrix rows. They are widely used as importance sampling probabilities in randomized matrix algorithms for problems ranging from regression (Cohen et al., 2015; Alaoui & Mahoney, 2015), to low-rank approximation (Drineas et al., 2008; Cohen et al., 2017; Musco & Musco, 2017; Rudi et al., 2018), to graph sparsification (Spielman & Srivastava, 2011), and beyond (Agarwal et al., 2020).

More recently, leverage score sampling has been applied extensively in active learning as a method for selecting data examples for training (Cohen & Migliorati, 2017; Avron et al., 2019; Chen & Price, 2019; Erdélyi et al., 2020; Chen & Derezinski, 2021; Gajjar et al., 2023; Musco et al.; Cardenas et al., 2023; Shimizu et al., 2024). The efficient estimation of Shapley values via the regression problem in Equation 4 can be viewed as an active learning problem, since our primary cost is in observing entries in the target vector $\mathbf{y}$, each of which corresponds to an expensive value function evaluation.

For active learning of linear models with $n$ features under the $\ell_2$ norm, the $O(n \log n + n/\epsilon)$ sample complexity required by leverage score sampling is known to be near-optimal. More complicated sampling methods can achieve $O(n/\epsilon)$, which is optimal in the worst-case (Batson et al., 2012; Chen & Price, 2019). However, it is not clear how to apply such methods efficiently to an exponentially tall matrix, as we manage to do for leverage score sampling.

## 2 BACKGROUND AND NOTATION

**Notation.** Lowercase letters represent scalars, bold lowercase letters vectors, and bold uppercase letters matrices. We use the set notation $[n] = \{1, \ldots, n\}$ and $\emptyset = \{\}$. We let $\mathbf{0}$ denote the all zeros vector, $\mathbf{1}$ the all ones vector, and $\mathbf{I}$ the identity matrix, with dimensions clear from context. For a vector $\mathbf{x}$, $x_i$ is the $i^{th}$ entry (non-bold to indicate a scalar). For a matrix $\mathbf{X} \in \mathbb{R}^{\rho \times n}$, $[\mathbf{X}]_i \in \mathbb{R}^{1 \times n}$ is the $i^{th}$ row. For a vector $\mathbf{x}$, $\|\mathbf{x}\|_2 = (\sum_i x_i^2)^{1/2}$ denotes the Euclidean ($\ell_2$) norm and $\|\mathbf{x}\|_1 = \sum_i |x_i|$ is the $\ell_1$ norm. For a matrix $\mathbf{X} \in \mathbb{R}^{n \times m}$, $\mathbf{X}^+$ denotes the Moore–Penrose pseudoinverse.

**Preliminaries.** Any subset $S \subseteq [n]$ can be represented by a binary indicator vector $\mathbf{z} \in \{0, 1\}^n$, and we use use $v(S)$ and $v(\mathbf{z})$ interchangeably. We construct the matrix $\mathbf{Z}$ and target vector $\mathbf{y}$ appearing in Equation 2 by indexing rows by all $\mathbf{z} \in \{0, 1\}^n$ with $0 < \|\mathbf{z}\|_1 < n$:

- Let $\mathbf{Z} \in \mathbb{R}^{2^n - 2 \times n}$ be a matrix with $[\mathbf{Z}]_{\mathbf{z}} = \sqrt{w(\|\mathbf{z}\|_1)} \mathbf{z}^\top$.
- Let $\mathbf{y} \in \mathbb{R}^{2^n - 2}$ be the vector where $[\mathbf{y}]_{\mathbf{z}} = \sqrt{w(\|\mathbf{z}\|_1)}(v(\mathbf{z}) - v(\mathbf{0}))$.

Above, $w(s) = \left(\binom{n}{s} s(n - s)\right)^{-1}$ is the same weight function defined in Equation 4.

As discussed, the Kernel SHAP method is based on an equivalence between Shapley values and the solution of a constrained regression problem involving $\mathbf{Z}$ and $\mathbf{y}$. Formally, we have:

**Lemma 2.1** (Equivalence (Lundberg & Lee, 2017; Charnes et al., 1988))**.**

$$\phi = \underset{\mathbf{x}:\langle \mathbf{x}, \mathbf{1}\rangle = v(\mathbf{1}) - v(\mathbf{0})}{\arg\min} \|\mathbf{Z}\mathbf{x} - \mathbf{y}\|_2^2 \tag{6}$$

$$= \underset{\mathbf{x}:\langle \mathbf{x}, \mathbf{1}\rangle = v(\mathbf{1}) - v(\mathbf{0})}{\arg\min} \sum_{\mathbf{z}:0<\|\mathbf{z}\|_1<n} w(\|\mathbf{z}\|_1) \cdot (\langle \mathbf{z}, \mathbf{x}\rangle - (v(\mathbf{z}) - v(\mathbf{0})))^2. \tag{7}$$

For completeness, we provide a self-contained proof of Lemma 2.1 in Appendix I. The form in Equation 7 inspires the heuristic choice to sample sets with probabilities proportional to $w(\|\mathbf{z}\|_1)$ in Kernel SHAP, as larger terms in the sum should intuitively be sampled with higher probability to reduce variance of the estimate. However, as discussed, a more principled way to approximately solve least squares regression problems via subsampling is to use probabilities proportional to the statistical leverage scores. Formally, these scores are defined as follows:

**Definition 2.2** (Leverage Scores)**.** *Consider a matrix* $\mathbf{X} \in \mathbb{R}^{\rho \times n}$. *For* $i \in [\rho]$, *the leverage score of the* $i^{th}$ *row* $[\mathbf{X}]_i \in \mathbb{R}^{1 \times n}$ *is* $\ell_i := [\mathbf{X}]_i (\mathbf{X}^\top \mathbf{X})^+ [\mathbf{X}]_i^\top$.

## 3 LEVERAGE SHAP

Statistical leverage scores are traditionally used to approximately solve unconstrained regression problems. Since Shapley values are the solution to a linearly constrained problem, we first reformulate this into an unconstrained problem. Ultimately, we sample by leverage scores of the reformulated problem, which we prove have a simple closed form that admits efficient sampling.

Concretely, we have the following equivalence:

**Lemma 3.1** (Constrained to Unconstrained). *Let $\mathbf{P}$ be the projection matrix $\mathbf{I} - \frac{1}{n}\mathbf{1}\mathbf{1}^\top$. Define $\mathbf{A} = \mathbf{Z}\mathbf{P}$ and $\mathbf{b} = \mathbf{y} - \mathbf{Z}\mathbf{1}\frac{v(\mathbf{1})-v(\mathbf{0})}{n}$. Then*

$$\underset{\mathbf{x}:\langle\mathbf{x},\mathbf{1}\rangle=v(\mathbf{1})-v(\mathbf{0})}{\arg\min} \|\mathbf{Z}\mathbf{x} - \mathbf{y}\|_2^2 = \arg\min_{\mathbf{x}} \|\mathbf{A}\mathbf{x} - \mathbf{b}\|_2^2 + \mathbf{1}\frac{v(\mathbf{1}) - v(\mathbf{0})}{n}. \tag{8}$$

*Further, we have that* $\min_{\mathbf{x}:\langle\mathbf{x},\mathbf{1}\rangle=v(\mathbf{1})-v(\mathbf{0})} \|\mathbf{Z}\mathbf{x} - \mathbf{y}\|_2^2 = \min_{\mathbf{x}} \|\mathbf{A}\mathbf{x} - \mathbf{b}\|_2^2$.

When there are multiple $\mathbf{x}$ that minimize the objective $\|\mathbf{A}\mathbf{x} - \mathbf{b}\|_2^2$, we define $\arg\min_{\mathbf{x}} \|\mathbf{A}\mathbf{x} - \mathbf{b}\|_2^2$ as the $\mathbf{x}$ that also minimizes $\|\mathbf{x}\|_2^2$.

Lemma 3.1 is proven in Appendix K. Using the equivalent unconstrained regressio n problem, we consider methods that construct a sampling matrix $\mathbf{S}$ and return:

$$\tilde{\phi} = \arg\min_{\mathbf{x}} \|\mathbf{S}\mathbf{A}\mathbf{x} - \mathbf{S}\mathbf{b}\|_2^2 + \mathbf{1}\frac{v(\mathbf{1}) - v(\mathbf{0})}{n}. \tag{9}$$

The main question is how to build $\mathbf{S}$. Our Leverage SHAP method does so by sampling with probabilities proportional to the leverage scores of $\mathbf{A}$. Since naively these scores would be intractable to compute, requiring $O(\rho n^2)$ time, where $\rho = 2^n - 2$ is the number of rows in $\mathbf{A}$, our method rests on the derivation of a simple closed form expression for the leverage scores, which we prove below.

### 3.1 ANALYTICAL FORM OF LEVERAGE SCORES

**Lemma 3.2.** *Let $\mathbf{A}$ be as defined in Lemma 3.1. The leverage score of the row in $\mathbf{A}$ with index $\mathbf{z} \in \{0,1\}^n$, where $0 < \|\mathbf{z}\|_1 < n$, is equal to $\ell_{\mathbf{z}} = \binom{n}{\|\mathbf{z}\|_1}^{-1}$.*

The proof of Lemma 3.2 depends on an explicit expression for the matrix $\mathbf{A}^\top\mathbf{A}$:

**Lemma 3.3.** *Let $\mathbf{A}$ be as defined in Lemma 3.1. $\mathbf{A}^\top\mathbf{A} = \frac{1}{n}\mathbf{P}$.*

This fact will also be relevant later in our analysis, as it implies that $\mathbf{A}^\top\mathbf{A}$ is well-conditioned. Perhaps surprisingly, all of its non-zero singular values are equal to $1/n$.

*Proof of Lemma 3.3.* Recall that $\mathbf{A} = \mathbf{Z}\mathbf{P}$, so $\mathbf{A}^\top\mathbf{A} = \mathbf{P}^\top\mathbf{Z}^\top\mathbf{Z}\mathbf{P}$. We start by deriving explicit expressions for the $(i,j)$ entry of $\mathbf{Z}^\top\mathbf{Z}$, denoted by $[\mathbf{Z}^\top\mathbf{Z}]_{(i,j)}$, for all $i \in [n]$ and $j \in [n]$. In particular, we can check that $[\mathbf{Z}^\top\mathbf{Z}]_{(i,j)} = \sum_{\mathbf{z}\in\{0,1\}^n} z_i z_j w(\|\mathbf{z}\|_1)$. So, when $i \neq j$,

$$[\mathbf{Z}^\top\mathbf{Z}]_{(i,j)} = \sum_{\mathbf{z}:z_i,z_j=1} w(\|\mathbf{z}\|_1) = \sum_{s=2}^{n-1} \binom{n-2}{s-2} w(s). \tag{10}$$

Let $c_n$ denote the above quantity, which does not depend on $i$ and $j$. I.e., $[\mathbf{Z}^\top\mathbf{Z}]_{(i,j)} = c_n$ for $i \neq j$.

For the case $i = j$, let $j'$ be any fixed index not equal to $i$. We have that

$$[\mathbf{Z}^\top\mathbf{Z}]_{(i,i)} = \sum_{\mathbf{z}:z_i=1} w(\|\mathbf{z}\|_1) = \sum_{\mathbf{z}:z_i=1,z_{j'}=0} w(\|\mathbf{z}\|_1) + \sum_{\mathbf{z}:z_i=1,z_{j'}=1} w(\|\mathbf{z}\|_1)$$

$$= \sum_{s=1}^{n-1} \binom{n-2}{s-1} w(s) + \sum_{s=2}^{n-1} \binom{n-2}{s-2} w(s) = \frac{1}{n} + c_n. \tag{11}$$

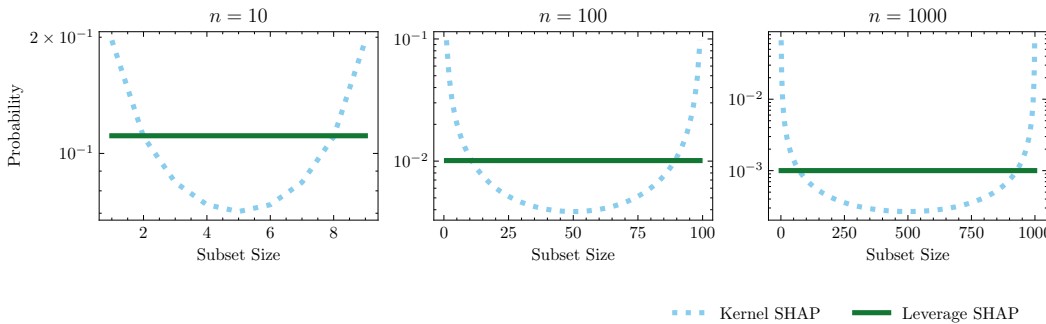

Figure 2: Let $S$ be a subset sampled with the Kernel SHAP or Leverage SHAP probabilities. The plots show the distribution of the set size $|S|$ for different $n$. As in in the definition of Shapley values, the leverage score distribution places equal *total weight* on each subset size, contrasting with Kernel SHAP which over-weights small and large subsets.

The last equality can be verified via a direct calculation. By Equations 10 and 11, we have that $\mathbf{Z}^\top \mathbf{Z} = \frac{1}{n}\mathbf{I} + c_n \mathbf{1}\mathbf{1}^\top$. So we conclude that:

$$\mathbf{A}^\top \mathbf{A} = \mathbf{P}^\top \mathbf{Z}^\top \mathbf{Z} \mathbf{P} = \left(\mathbf{I} - \frac{1}{n}\mathbf{1}\mathbf{1}^\top\right)\left(\frac{1}{n}\mathbf{I} + c_n \mathbf{1}\mathbf{1}^\top\right)\left(\mathbf{I} - \frac{1}{n}\mathbf{1}\mathbf{1}^\top\right) = \frac{1}{n}\left(\mathbf{I} - \frac{1}{n}\mathbf{1}\mathbf{1}^\top\right). \quad \square$$

With Lemma 3.3, we are now ready to analytically compute the leverage scores.

*Proof of Lemma 3.2.* The leverage scores of $\mathbf{A}$ are given by $\ell_{\mathbf{z}} = [\mathbf{ZP}]_{\mathbf{z}} \left(\mathbf{A}^\top \mathbf{A}\right)^+ [\mathbf{ZP}]_{\mathbf{z}}^\top$. By Lemma 3.3, we have that $\mathbf{A}^\top \mathbf{A} = \frac{1}{n}\left(\mathbf{I} - \frac{1}{n}\mathbf{1}\mathbf{1}^\top\right)$ and thus $(\mathbf{A}^\top \mathbf{A})^+ = n\mathbf{I} - \mathbf{1}\mathbf{1}^\top$. Recall that $[\mathbf{Z}]_{\mathbf{z}} = \sqrt{w(\|\mathbf{z}\|_1)}\mathbf{z}^\top$ so $[\mathbf{ZP}]_{\mathbf{z}} = \sqrt{w(\|\mathbf{z}\|_1)}\mathbf{z}^\top\left(\mathbf{I} - \frac{1}{n}\mathbf{1}\mathbf{1}^\top\right)$. We thus have:

$$\ell_{\mathbf{z}} = \sqrt{w(\|\mathbf{z}\|_1)}\left(\mathbf{z} - \frac{\|\mathbf{z}\|_1}{n}\mathbf{1}\right)^\top (n\mathbf{I} - \mathbf{1}\mathbf{1}^\top)\sqrt{w(\|\mathbf{z}\|_1)}\left(\mathbf{z} - \frac{\|\mathbf{z}\|_1}{n}\mathbf{1}\right)$$

$$= w(\|\mathbf{z}\|_1)(n\|\mathbf{z}\|_1 - \|\mathbf{z}\|_1^2) = w(\|\mathbf{z}\|_1)\|\mathbf{z}\|_1(n - \|\mathbf{z}\|_1) = \binom{n}{\|\mathbf{z}\|_1}^{-1}. \quad \square$$

Notice that the leverage scores simply correspond to sampling every row proportional to the number of sets of that size. In retrospect, this is quite intuitive given the original definition of Shapley values in Equation 1. As shown in Figure 2, unlike the Kernel SHAP sampling distribution, leverage score sampling ensures that all subset sizes are equally represented in our sample from $\mathbf{A}$.

## 3.2 OUR ALGORITHM

With Lemma 3.2 in place, we are ready to present Leverage SHAP, which is given as Algorithm 1. In addition to leverage score sampling, the algorithm incorporates paired sampling and sampling without replacement, both of which are used in optimized implementations of Kernel SHAP.

For paired sampling, the idea is to sample rows with probabilities proportional to the leverage scores, but in a *correlated way*: any time we sample index $\mathbf{z}$, we also select its complement, $\bar{\mathbf{z}}$, where $\bar{z}_i = 1 - z_i$ for $i \in [n]$. Note that, by the symmetry of $\mathbf{A}$'s leverage scores, $\ell_{\mathbf{z}} = \ell_{\bar{\mathbf{z}}}$. Similarly, $w(\|\mathbf{z}\|_1) = w(\|\bar{\mathbf{z}}\|_1)$. Moving forward, let $\mathcal{Z}$ denote the set of pairs $(\mathbf{z}, \bar{\mathbf{z}})$ where $0 < \|\mathbf{z}\|_1 < n$.

To perform paired sampling without replacement, we select indices $(\mathbf{z}, \bar{\mathbf{z}})$ independently at random with probability $\min(1, c(\ell_{\mathbf{z}} + \ell_{\bar{\mathbf{z}}})) = \min(1, 2c\ell_{\mathbf{z}})$, where $c > 1$ is an oversampling parameter. All rows that are sampled are included in a subsampled matrix $\mathbf{Z}'\mathbf{P}$ and reweighted by the inverse of the probability with which they were sampled. The expected number of row samples in $\mathbf{Z}'\mathbf{P}$ is equal to $\sum_{(\mathbf{z}, \bar{\mathbf{z}})} \min(1, 2c\ell_{\mathbf{z}})$. We choose $c$ via binary search so that this expectation equals $m - 2$, where $m$ is our target number of value function evaluations (two evaluations are reserved to compute

---

**Algorithm 1** Leverage SHAP

---

**Input:** $n$: number of players, $v$: set function, $m$: target number of function evaluations (at least $n$)
**Output:** $\tilde{\phi}$: approximate Shapley values
  1: $m \leftarrow \min(m, 2^n)$                                     ▷ $2^n$ samples to solve exactly
  2: Find $c$ via binary search so that $m - 2 = \sum_{s=1}^{\lfloor n/2 \rfloor} \min(\binom{n}{s}, 2c)$         ▷ Equation 12
  3: $\mathcal{Z}' \leftarrow \texttt{BernoulliSample}(n, c)$     ▷ Sample from $\mathcal{Z}$ without replacement (Algorithm 2)
  4: $i \leftarrow 0$, $\mathbf{W} \leftarrow \mathbf{0}_{2|\mathcal{Z}'| \times 2|\mathcal{Z}'|}$, $\mathbf{Z}' \leftarrow \mathbf{0}_{2|\mathcal{Z}'| \times n}$
  5: **for** $(\mathbf{z}, \bar{\mathbf{z}}) \in \mathcal{Z}'$ **do**                             ▷ Build sampled regression problem
  6:      $\mathbf{Z}'_{(i,)} \leftarrow \mathbf{z}^\top$, $\mathbf{Z}'_{(i+1,)} \leftarrow \bar{\mathbf{z}}^\top$
  7:      $W_{(i,i)} \leftarrow W_{(i+1,i+1)} \leftarrow \frac{w(\|\mathbf{z}\|_1)}{\min(1, 2c \cdot \ell_{\mathbf{z}})}$         ▷ Correct weights in expectation
  8:      $i \leftarrow i + 2$
  9: **end for**
10: $\mathbf{y}' \leftarrow v(\mathbf{Z}') - v(\mathbf{0})\mathbf{1}$                        ▷ $v(\mathbf{Z}')$ evaluates $v$ on $m - 2$ inputs
11: $\mathbf{b}' \leftarrow \mathbf{y}' - \frac{v(\mathbf{1}) - v(\mathbf{0})}{n}\mathbf{Z}'\mathbf{1}$
12: Compute

$$\tilde{\phi}^\perp \leftarrow \arg\min_{\mathbf{x}} \|\mathbf{W}^{\frac{1}{2}}\mathbf{Z}'\mathbf{P}\mathbf{x} - \mathbf{W}^{\frac{1}{2}}\mathbf{b}'\|_2$$

                                                             ▷ Standard least squares

13: **return** $\tilde{\phi} \leftarrow \tilde{\phi}^\perp + \frac{v(\mathbf{1}) - v(\mathbf{0})}{n}\mathbf{1}$

---

$v(\mathbf{1}) - v(\mathbf{0})$). I.e., we choose $c$ to solve the equation:

$$m - 2 = \sum_{s=1}^{n-1} \min\left(1, 2c\binom{n}{s}^{-1}\right)\binom{n}{s}. \tag{12}$$

Note that our procedure is different from the with-replacement Kernel SHAP procedure described in Section 1.1. Sampling without replacement requires more care to avoid iterating over the exponential number of pairs in $\mathcal{Z}$. Fortunately, we can exploit the fact that there are only $n-1$ different set sizes, and all sets of the same size have the same leverage score. This allows us to determine how many sets of a different size should be sampled (by drawing a binomial random variable), and we can then sample the required number of sets of each size uniformly at random. Ultimately, we can collect $m$ samples in $O(mn^2)$ time. Details are deferred to Appendix J.

Because we sample independently without replacement, it is possible that the number of function evaluations $|\mathcal{Z}'| + 2$ used by Algorithm 1 exceeds $m$. However, because this number is itself a sum of independent random variables, it tightly concentrates around its expectation, $m$. If necessary, Leverage SHAP can be implemented so that the number of evaluations is deterministically $m$ (i.e., instead of sampling from a Binomial on Line 4 in Algorithm 2, set $m_s$ to be the expectation of the Binomial). Since this version of Leverage SHAP no longer has independence, our theoretical analysis no longer applies. However, we still believe the guarantees hold even without independence, and leave a formal analysis to future work.

## 4 THEORETICAL GUARANTEES

As discussed, Leverage SHAP offers strong theoretical approximation guarantees. Our main result is Theorem 1.1 where we show that, with $m = O(n \log(n/\delta) + \frac{n}{\epsilon\delta})$ samples in expectation, Algorithm 1 returns a solution $\tilde{\phi}$ that, with probability $1 - \delta$, satisfies

$$\|\mathbf{A}\tilde{\phi} - \mathbf{b}\|_2^2 \leq (1 + \epsilon)\|\mathbf{A}\phi - \mathbf{b}\|_2^2. \tag{13}$$

We prove this guarantee in Appendix A. The proof modifies the standard analysis of leverage scores for active least squares regression, which requires two components: 1) a *subspace embedding guarantee*, proven with a matrix Chernoff bound applied to a sum of rank-1 random matrices, and 2) an *approximate matrix-multiplication guarantee*, proven with a second moment analysis. We replace these steps with 1) a matrix Bernstein bound applied to a sum of rank-2 random matrices and 2) a block-approximate matrix multiplication guarantee.

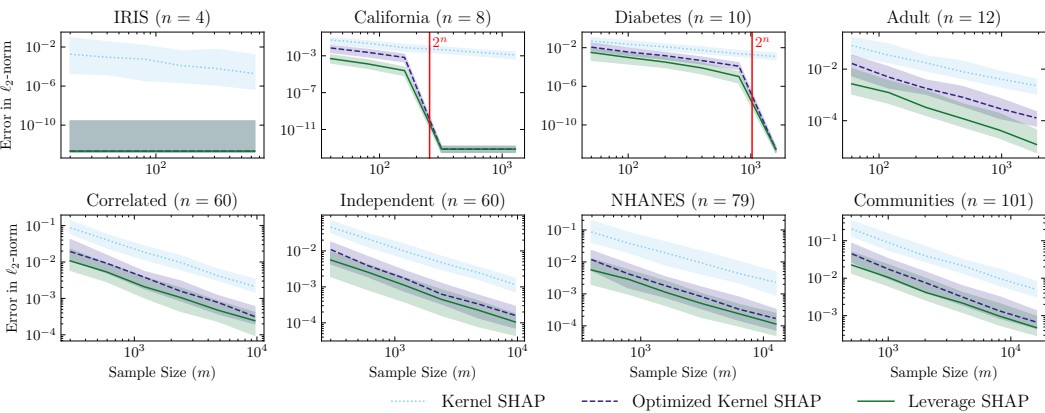

Figure 3: The $\ell_2$-norm error between the estimated Shapley values and ground truth Shapley values as a function of sample size. The lines report the median error while the shaded regions encompass the first to third quartile over 100 runs. Besides the setting where the exact Shapley values can be recovered, Leverage SHAP reliably outperforms even Optimized Kernel SHAP: the third quartile of the Leverage SHAP error is roughly the median error of Optimized Kernel SHAP.

While well-motivated by the connection between Shapley values and linear regression, the objective function in Theorem 1.1 is not intuitive. Instead, we may be interested in the $\ell_2$-norm error between the true Shapley values and the estimated Shapley values. Fortunately, we can use special properties of our problem to show that Theorem 1.1 implies the following corollary on the $\ell_2$-norm error.

**Corollary 4.1.** *Suppose* $\tilde{\phi}$ *satisfies* $\|\mathbf{A}\tilde{\phi}-\mathbf{b}\|_2^2 \leq (1+\epsilon)\|\mathbf{A}\phi - \mathbf{b}\|_2^2$. *Let* $\gamma = \|\mathbf{A}\phi-\mathbf{b}\|_2^2/\|\mathbf{A}\phi\|_2^2$. *Then*

$$\|\phi - \tilde{\phi}\|_2^2 \leq \epsilon\gamma\|\phi\|_2^2.$$

The proof of Corollary 4.1 appears in Appendix B. The statement of the corollary is in terms of a problem-specific parameter $\gamma$ which intuitively measures how well the *optimal* coefficients $\phi$ can reach the target vector $\mathbf{b}$. In Appendix F, we explore how the performance varies with $\gamma$ in practice. We find (see e.g., Figures 5, 6, 7) that performance does erode as $\gamma$ increases for all regression-based algorithms, suggesting that $\gamma$ is not an artifact of our analysis.

## 5 EXPERIMENTS

In the experiments, we evaluate Leverage SHAP and Kernel SHAP based on how closely they align with the ground truth Shapley values. We primarily use the $\ell_2$-norm error i.e., $\|\phi - \tilde{\phi}\|_2^2/\|\phi\|_2^2$ as the error metric. We run our experiments on eight popular datasets from the SHAP library (Lundberg & Lee, 2017). We find that Leverage SHAP outperforms the highly optimized Kernel SHAP, achieving a 50% reduction in error on average (see Table 2). In addition, we find that Leverage SHAP consistently outperforms Permutation SHAP (see the additional experiments in Section G).

**Implementation Details.** In order to compute the ground truth Shapley values for large values of $n$, we use a tree-based model for which we can compute the exact Shapley values efficiently using Tree SHAP (Lundberg & Lee, 2017). All of our code is written in Python and can be found on Github[3]. We use the SHAP library for the Optimized Kernel SHAP implementation, Tree SHAP, and the datasets. We use XGBoost for training and evaluating trees, under the default parameters.

**Additional experiments.** So that we do not overcrowd the plots, we show performance on Kernel SHAP, the Optimized Kernel SHAP implementation, and Leverage SHAP in the main body. We also run all of the experiments with the ablated estimators in Appendix G. The ablation experiments (see e.g., Figure 8) suggest that Bernoulli sampling (sampling without replacement) improves Leverage SHAP for small $n$ and the paired sampling improves Leverage SHAP for all settings. Because it is more interpretable, we report the $\ell_2$-norm error metric to compare the estimated to the ground truth Shapley values. We also report the linear objective error for similar experiments in Appendix H.

---

[3]github.com/rtealwitter/leverageshap

| | IRIS | California | Diabetes | Adult | Correlated | Independent | NHANES | Communities |
|---|---|---|---|---|---|---|---|---|
| **Kernel SHAP** | | | | | | | | |
| Mean | 0.026 | 0.0266 | 0.0553 | 0.0673 | 0.0465 | 0.0264 | 0.0604 | 0.12 |
| 1st Quartile | 1.61e-05 | 0.00829 | 0.0116 | 0.0182 | 0.0244 | 0.0134 | 0.0202 | 0.0563 |
| 2nd Quartile | 0.000898 | 0.0236 | 0.0229 | 0.0345 | 0.0404 | 0.0217 | 0.0388 | 0.089 |
| 3rd Quartile | 0.0328 | 0.0424 | 0.0524 | 0.0936 | 0.056 | 0.0303 | 0.0843 | 0.149 |
| **Optimized Kernel SHAP** | | | | | | | | |
| Mean | 4.84e-09 | 0.00342 | 0.0093 | 0.00989 | 0.0117 | 0.00474 | 0.00758 | 0.0233 |
| 1st Quartile | 1.66e-13 | 0.000802 | 0.00161 | 0.00187 | 0.00499 | 0.00194 | 0.00156 | 0.00962 |
| 2nd Quartile | 2.17e-13 | 0.00238 | 0.00356 | 0.00489 | 0.00916 | 0.00391 | 0.00425 | 0.0173 |
| 3rd Quartile | 2.69e-10 | 0.00489 | 0.00868 | 0.0122 | 0.015 | 0.00695 | 0.00871 | 0.0325 |
| **Leverage SHAP** | | | | | | | | |
| Mean | 4.84e-09 | 0.000311 | 0.0023 | 0.00477 | 0.00716 | 0.00288 | 0.00532 | 0.0156 |
| 1st Quartile | 1.66e-13 | 4.47e-05 | 0.000215 | 0.000477 | 0.00289 | 0.000843 | 0.000995 | 0.0062 |
| 2nd Quartile | 2.17e-13 | 0.000133 | 0.000969 | 0.00124 | 0.00528 | 0.00257 | 0.00288 | 0.0104 |
| 3rd Quartile | 2.69e-10 | 0.000366 | 0.00241 | 0.00354 | 0.00891 | 0.00417 | 0.00554 | 0.0225 |

Table 1: Summary statistics of the $\ell_2$-norm error for every dataset. We adopt the Olympic medal convention: gold , silver and bronze cells signify first, second and third best performance, respectively. Except for ties, Leverage SHAP gives the best performance across all settings.

Figure 3 plots the performance for Kernel SHAP, the Optimized Kernel SHAP, and Leverage SHAP as the number of samples varies (we set $m = 5n, 10n, 20n, \dots, 160n$). For each estimator, the line is the median error and the shaded region encompasses the first and third quartile over 100 random runs.[4] Kernel SHAP gives the highest error in all settings of $m$ and $n$, pointing to the importance of the paired sampling and sampling without replacement optimizations in both Optimized Kernel SHAP and Leverage SHAP. Because both methods sample without replacement, Optimized Kernel SHAP and Leverage SHAP achieve essentially machine precision as $m$ approaches $2^n$. For all other settings, Leverage SHAP reliably outperforms Optimized Kernel SHAP; the third quartile of Leverage SHAP is generally at the median of Optimized Kernel SHAP. The analogous experiment for the linear objective error depicted in Figure 10 indicates the same findings.

Table 1 depicts the estimator performance for $m = 10n$. The table shows the mean, first quartile, second quartile, and third quartile of the error for each estimator on every dataset. Except for ties in the setting where $m \geq 2^n$, Leverage SHAP gives the best performance followed by Optimized Kernel SHAP and then Kernel SHAP.

Beyond sample size, we evaluate how the estimators perform with noisy access to the set functions in Appendix D. As in the sample size experiments, Leverage SHAP meets or exceeds the other estimators, suggesting the utility of the algorithm as a robust estimator in explainable AI.

# 6 CONCLUSION

We introduce Leverage SHAP, a principled alternative to Kernel SHAP, designed to provide provable accuracy guarantees with nearly linear model evaluations. The cornerstone of our approach is leverage score sampling, a powerful subsampling technique used in regression. To adapt this method for estimating Shapley values, we reformulate the standard Shapley value constrained regression problem and analytically compute the leverage scores of this reformulation. Leverage SHAP efficiently uses these scores to produce provably accurate estimates of Shapley values. Our method enjoys strong theoretical guarantees, which we prove by modifying the standard leverage score analysis to incorporate the empirically-motivated paired sampling and sampling without replacement optimizations. Through extensive experiments on eight datasets, we demonstrate that our algorithm outperforms even the optimized version of Kernel SHAP, establishing Leverage SHAP as a valuable tool in the explainable AI toolkit.

---

[4]We report the median and quartiles instead of the mean and standard deviation since the mean minus standard deviation is negative for the small error values in our experiments.

## 7 ACKNOWLEDGEMENTS

This work was supported by the National Science Foundation under Grant No. 2045590 and Graduate Research Fellowship Grant No. DGE-2234660. We are grateful to Yurong Liu for introducing us to the problem of estimating Shapley values.

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

# Appendices

## CONTENTS

.

## A  PROOF OF APPROXIMATION GUARANTEE

In this section, we prove the following theorem.

**Theorem A.1.** *Let $m = O(n \log(\frac{n}{\delta}) + n \frac{1}{\delta \epsilon})$. Algorithm 1 produces an estimate $\tilde{\phi}$ such that, with probability $1 - \delta$,*

$$\|\mathbf{A}\tilde{\phi} - \mathbf{b}\|_2^2 \le (1 + \epsilon)\|\mathbf{A}\phi - \mathbf{b}\|_2^2.$$

Because of the connection between the constrained and unconstrained regression problems described in Lemma 3.1, the theorem implies the theoretical guarantees in Theorem 1.1. The time complexity bound in Theorem 1.1 follows from Lemma J.1 in Appendix J.

Consider the following leverage score sampling scheme where rows are selected in blocks. (We use the word "block" instead of "pair" in the formal analysis for generality.) Let $\Theta$ be a partition into blocks of equal size (size 2 in our case) where the leverage scores within a block are equal. Block $\Theta_i$ is independently sampled with probability $p_i^+ := \min(1, c \sum_{k \in \Theta_i} \ell_k)$ for a constant $c$. The constant $c$ is chosen so that the expected number of blocks sampled is $m$, i.e., $\sum_i p_i^+ = m$. Let $m'$ be the number of blocks sampled. Let $\mathbf{S} \in \mathbb{R}^{|\Theta_i|m' \times \rho}$ be the (random) sampling matrix. Each row of $\mathbf{S}$ corresponds to a row $k$ from a sampled block $i$: every entry in the row is 0 except for the $k$th entry which is $1/\sqrt{p_i^+}$.

In order to analyze the solution returned by Algorithm 1, we will prove that the sampling matrix $\mathbf{S}$ preserves the spectral norm and Frobenius norm.

**Lemma A.2** (Bernoulli Spectral Approximation). *Let $\mathbf{U} \in \mathbb{R}^{\rho \times n}$ be a matrix with orthonormal columns. Consider the block random sampling matrix $\mathbf{S}$ described above with rows sampled according to the leverage scores of $\mathbf{U}$. When $m = \Omega(n \log(n/\delta)/\epsilon^2)$,*

$$\|\mathbf{I} - \mathbf{U}^\top \mathbf{S}^\top \mathbf{S} \mathbf{U}\|_2 \le \epsilon \tag{14}$$

*with probability $1 - \delta$.*

*Proof of Lemma A.2.* We will use the following matrix Bernstein bound (see e.g., Theorem 6.6.1 in Tropp et al. (2015)).

**Fact A.3** (Matrix Bernstein). *Consider a finite sequence $\{\mathbf{X}_i\}$ of independent, random, Hermitian matrices with dimension $n$. Assume that $\mathbb{E}[\mathbf{X}_k] = \mathbf{0}$ and $\|\mathbf{X}_i\|_2 \le L$ for all $i$. Define $\mathbf{X} = \sum_i \mathbf{X}_i$ and let $V = \|\mathbb{E}[\mathbf{X}^2]\|_2 = \|\sum_i \mathbb{E}[\mathbf{X}_i^2]\|_2$. Then for any $\epsilon > 0$,*

$$\Pr\left(\|\mathbf{X}\|_2 \ge \epsilon\right) \le n \exp\left(\frac{-\epsilon^2/2}{V + L\epsilon/3}\right). \tag{15}$$

Let $\mathbf{U}_{(k,)} \in \mathbb{R}^{1 \times n}$ be the $k$th row of $\mathbf{U}$. Similarly, let $\mathbf{U}_{\Theta_i} \in \mathbb{R}^{|\theta_i| \times n}$ be the matrix with rows $\mathbf{U}_{(k,)}$ for $k \in \Theta_i$.

We will choose $\mathbf{X}_i = \mathbf{U}_{\Theta_i}^\top \mathbf{U}_{\Theta_i} - \frac{1}{p_i^+} \mathbf{U}_{\Theta_i}^\top \mathbf{U}_{\Theta_i} \mathbb{1}[i \text{ selected}]$. Then $\mathbb{E}[\mathbf{X}_i] = \mathbf{0}$ and $\mathbf{X} = \mathbf{I} - \mathbf{U}^\top \mathbf{S}^\top \mathbf{S} \mathbf{U}$.

First, we will upper bound $\max_i \|\mathbf{X}_i\|_2$. If $i$ is not selected or $p_i^+ = 1$, then $\|\mathbf{X}_i\|_2 = 0$ so we only need to consider the case where $i$ is selected and $p_i^+ = c \sum_{k \in \Theta_i} \ell_k < 1$. Then

$$\|\mathbf{X}_i\|_2 \le \left|1 - \frac{1}{p_i^+}\right| \|\mathbf{U}_{\Theta_i}^\top \mathbf{U}_{\Theta_i}\|_2 \le \frac{1}{p_i^+} \sum_{k \in \Theta_i} \|\mathbf{U}_{(k,)}\|_2^2 = \frac{1}{c} = L \tag{16}$$

Next, we will upper bound $\|\sum_i \mathbb{E}[\mathbf{X}_i^2]\|_2$. Again, notice that $\mathbb{E}[\mathbf{X}_i^2] = \mathbf{0}$ if $p_i^+ = 1$ so we only need to consider the case where $p_i^+ < 1$. Then

$$\sum_i \mathbb{E}[\mathbf{X}_i^2] = \sum_{i:p_i^+<1} p_i^+ \left(1 - \frac{1}{p_i^+}\right)^2 \left(\mathbf{U}_{\Theta_i}^\top \mathbf{U}_{\Theta_i}\right)^2 + (1 - p_i^+)\left(\mathbf{U}_{\Theta_i}^\top \mathbf{U}_{\Theta_i}\right)^2. \tag{17}$$

Notice that $p_i^+(1 - 1/p_i^+)^2 + (1 - p_i^+) = p_i^+ - 2 + 1/p_i^+ + 1 - p_i^+ = 1/p_i^+ - 1 \leq 1/p_i^+$. Therefore

$$(17) \preceq \sum_{i:p_i^+<1} \frac{1}{p_i^+} \mathbf{U}_{\Theta_i}^\top \mathbf{U}_{\Theta_i} \mathbf{U}_{\Theta_i}^\top \mathbf{U}_{\Theta_i} \tag{18}$$

Observe that entry $(k, k')$ of $\mathbf{U}_{\Theta_i} \mathbf{U}_{\Theta_i}^\top \in \mathbb{R}^{|\Theta_i| \times |\Theta_i|}$ is $\mathbf{U}_k \mathbf{U}_{k'}^\top$. So the absolute value of each entry is $|\mathbf{U}_k \mathbf{U}_{k'}^\top| \leq \|\mathbf{U}_k\|_2 \|\mathbf{U}_{k'}\|_2 = \ell_k^{1/2} \ell_{k'}^{1/2}$ by Cauchy-Schwarz. Define $\ell_i^{\max} = \max_{k \in \Theta_i} \ell_k$ and $\ell_i^{\min} = \min_{k \in \Theta_i} \ell_k$. By the Gershgorin circle theorem, $\mathbf{U}_{\Theta_i} \mathbf{U}_{\Theta_i}^\top \preceq 2\ell_i^{\max}|\Theta_i|^2 \mathbf{I}$. Equivalently, $\mathbf{A} \preceq \mathbf{B}$, $\mathbf{x}^\top \mathbf{A} \mathbf{x} \leq \mathbf{x}^\top \mathbf{B} \mathbf{x}$ for all $\mathbf{x}$. Consider an arbitrary $\mathbf{z}$. We have $\mathbf{z}^\top \mathbf{C}^\top \mathbf{A} \mathbf{C} \mathbf{z} \leq \mathbf{z}^\top \mathbf{C}^\top \mathbf{B} \mathbf{C} \mathbf{z}$ since $\mathbf{C} \mathbf{z}$ is some $\mathbf{x}$. It follows that $\mathbf{U}_{\Theta_i}^\top \mathbf{U}_{\Theta_i} \mathbf{U}_{\Theta_i}^\top \mathbf{U}_{\Theta_i} \preceq 2\ell_i^{\max}|\Theta_i|^2 \mathbf{U}_{\Theta_i}^\top \mathbf{U}_{\Theta_i}$. Then

$$(18) \preceq \frac{1}{c} \sum_{i=1}^{|\Theta_i|} \frac{2\ell_i^{\max}|\Theta_i|^2 \mathbf{U}_{\Theta_i}^\top \mathbf{U}_{\Theta_i}}{\sum_{k \in \Theta_i} \ell_k} \preceq \frac{1}{c} \max_i 2|\Theta_i| \frac{\ell_i^{\max}}{\ell_i^{\min}} \mathbf{I}. \tag{19}$$

Since $|\Theta_i| \leq 2$ and the leverage scores in a block are all equal, $\|\mathbb{E}[\mathbf{X}_j^2]\|_2 \leq 4/c$.

Recall that $c$ is chosen so that $m = \sum_i \min(1, c\sum_{k \in \Theta_i} \ell_k) \leq c\sum_i \sum_{k \in \Theta_i} \ell_k = cn$, where the last equality follows because the leverage scores sum to $n$. So $1/c \leq n/m$.

Plugging in to Fact A.3, we have that

$$\Pr\left(\|\mathbf{I} - \mathbf{U}^\top \mathbf{S}^\top \mathbf{S} \mathbf{U}\|_2 \geq \epsilon\right) \leq n \exp\left(\frac{-m\epsilon^2/2n}{4 + \epsilon/3}\right) \leq \delta \tag{20}$$

provided $m = \Omega(n \log(n/\delta)/\epsilon^2)$.

$\square$

Next, we will prove that the sampling matrix preserves the Frobenius norm.

**Lemma A.4** (Frobenius Approximation). *Let $\mathbf{U} \in \mathbb{R}^{\rho \times n}$ be a matrix with orthonormal columns. Consider the block random sampling matrix $\mathbf{S}$ described above with rows sampled according to the leverage scores of $\mathbf{U}$. Let $\mathbf{V} \in \mathbb{R}^{\rho \times n'}$. As long as $m \geq \frac{1}{\delta\epsilon^2}$, then*

$$\left\|\mathbf{U}^\top \mathbf{S}^\top \mathbf{S} \mathbf{V} - \mathbf{U}^\top \mathbf{V}\right\|_F \leq \epsilon \|\mathbf{U}\|_F \|\mathbf{V}\|_F \tag{21}$$

*with probability $1 - \delta$.*

*Proof.* We adapt Proposition 2.2 in Wu (2018) to the setting where blocks are sampled without replacement.

$$\mathbb{E}[\|\mathbf{U}^\top \mathbf{S}^\top \mathbf{S} \mathbf{V} - \mathbf{U}^\top \mathbf{V}\|_F^2] = \sum_{h_1=1}^n \sum_{h_2=1}^n \mathbb{E}[(\mathbf{U}^\top \mathbf{S}^\top \mathbf{S} \mathbf{V} - \mathbf{U}^\top \mathbf{V})_{(h_1, h_2)}^2] \tag{22}$$

$$= \sum_{h_1=1}^n \sum_{h_2=1}^n \mathrm{Var}[(\mathbf{U}^\top \mathbf{S}^\top \mathbf{S} \mathbf{V})_{(h_1, h_2)}] \tag{23}$$

where the last equality follows because $\mathbb{E}[\mathbf{U}^\top \mathbf{S}^\top \mathbf{S} \mathbf{V}] = \mathbf{U}^\top \mathbf{V}$. We will consider an individual entry in terms of the blocks $i$. If $p_i^+ = 1$, then the entry is 0 so we only need to consider the case where $p_i^+ < 1$. Then, using independence, we have

$$\mathrm{Var}[(\mathbf{U}^\top \mathbf{S}^\top \mathbf{S} \mathbf{V})_{(h_1, h_2)}] = \sum_{i:p_i^+<1} \mathrm{Var}[\frac{1}{p_i^+} \mathbb{1}[i \text{ selected}](\mathbf{U}_{(h_1, \Theta_i)}^\top \mathbf{V}_{(\Theta_i, h_2)})] \tag{24}$$

$$\leq \sum_{i:p_i^+<1} \frac{1}{p_i^+} \left(\mathbf{U}_{(h_1, \Theta_i)}^\top \mathbf{V}_{(\Theta_i, h_2)}\right)^2 \tag{25}$$

so

$$(23) \leq \sum_{i:p_i^+ < 1} \|\mathbf{U}_{(,\Theta_i)}^\top \mathbf{V}_{(\Theta_i,)}\|_F^2 \leq \sum_{i:p_i^+ < 1} \frac{1}{p_i^+} \|\mathbf{U}_{(\Theta_i,)}\|_F^2 \|\mathbf{V}_{(\Theta_i,)}\|_F^2 \tag{26}$$

$$= \sum_{i:p_i^+ < 1} \frac{\sum_{k \in \Theta_i} \|\mathbf{U}_k\|_2^2}{c \sum_{k \in \Theta_i} \ell_k} \|\mathbf{V}_{(\Theta_i,)}\|_F^2 = \frac{1}{c} \|\mathbf{V}\|_F^2 \leq \frac{1}{m} \|\mathbf{U}\|_F^2 \|\mathbf{V}\|_F^2 \tag{27}$$

where the last equality follows because $m \leq cn$ and $\|\mathbf{U}\|_F^2 = n$.

By Markov's inequality,

$$\Pr\left(\left\|\mathbf{U}^\top \mathbf{S}^\top \mathbf{S} \mathbf{V} - \mathbf{U}^\top \mathbf{V}\right\|_F > \epsilon \|\mathbf{U}\|_F \|\mathbf{V}\|_F\right) \leq \frac{\mathbb{E}\left[\left\|\mathbf{U}^\top \mathbf{S}^\top \mathbf{S} \mathbf{V} - \mathbf{U}^\top \mathbf{V}\right\|_F^2\right]}{\epsilon^2 \|\mathbf{U}\|_F^2 \|\mathbf{V}\|_F^2} \leq \frac{1}{m\epsilon^2} \leq \delta \tag{28}$$

as long as $m \geq \frac{1}{\delta \epsilon^2}$. $\qquad\square$

With the spectral and Frobenius approximation guarantees from Lemmas A.2 and A.4, we are ready to prove Theorem A.1.

*Proof of Theorem A.1.* Observe that

$$\|\mathbf{A}\tilde{\phi} - \mathbf{b}\|_2^2 = \|\mathbf{A}\tilde{\phi} - \mathbf{A}\phi + \mathbf{A}\phi - \mathbf{b}\|_2^2 = \|\mathbf{A}\tilde{\phi} - \mathbf{A}\phi\|_2^2 + \|\mathbf{A}\phi - \mathbf{b}\|_2^2 \tag{29}$$

where the second equality follows because $\mathbf{A}\phi - \mathbf{b}$ is orthogonal to any vector in the span of $\mathbf{A}$. So to prove the theorem, it suffices to show that

$$\|\mathbf{A}\tilde{\phi} - \mathbf{A}\phi\|_2^2 \leq \epsilon \|\mathbf{A}\phi - \mathbf{b}\|_2^2. \tag{30}$$

Let $\mathbf{U} \in \mathbb{R}^{\rho \times n'}$ be an orthonormal matrix that spans the columns of $\mathbf{A}$ where $n' \leq n$. There is some $\mathbf{y}$ such that $\mathbf{U}\mathbf{y} = \mathbf{A}\phi$ and some $\tilde{\mathbf{y}}$ such that $\mathbf{U}\tilde{\mathbf{y}} = \mathbf{A}\tilde{\phi}$. Observe that $\|\mathbf{A}\tilde{\phi} - \mathbf{A}\phi\|_2 = \|\mathbf{U}\tilde{\mathbf{y}} - \mathbf{U}\mathbf{y}\|_2 = \|\tilde{\mathbf{y}} - \mathbf{y}\|_2$ where the last equality follows because $\mathbf{U}^\top \mathbf{U} = \mathbf{I}$.

By the reverse triangle inequality and the submultiplicavity of the spectral norm, we have

$$\|\tilde{\mathbf{y}} - \mathbf{y}\|_2 \leq \|\mathbf{U}^\top \mathbf{S}^\top \mathbf{S} \mathbf{U}(\tilde{\mathbf{y}} - \mathbf{y})\|_2 + \|\mathbf{U}^\top \mathbf{S}^\top \mathbf{S} \mathbf{U}(\tilde{\mathbf{y}} - \mathbf{y}) - (\tilde{\mathbf{y}} - \mathbf{y})\|_2 \tag{31}$$

$$\leq \|\mathbf{U}^\top \mathbf{S}^\top \mathbf{S} \mathbf{U}(\tilde{\mathbf{y}} - \mathbf{y})\|_2 + \|\mathbf{U}^\top \mathbf{S}^\top \mathbf{S} \mathbf{U} - \mathbf{I}\|_2 \|\tilde{\mathbf{y}} - \mathbf{y}\|_2. \tag{32}$$

Because $\mathbf{U}$ has the same leverage scores as $\mathbf{A}$ and the number of rows sampled in $\mathbf{S}$ is within a constant factor of $m$, we can apply Lemma A.2: With $m = O(n \log \frac{n}{\delta})$, we have $\|\mathbf{U}^\top \mathbf{S}^\top \mathbf{S} \mathbf{U} - \mathbf{I}\|_2 \leq \frac{1}{2}$ with probability $1 - \delta/2$. So, with probability $1 - \delta/2$,

$$\|\tilde{\mathbf{y}} - \mathbf{y}\|_2 \leq 2\|\mathbf{U}^\top \mathbf{S}^\top \mathbf{S} \mathbf{U}(\tilde{\mathbf{y}} - \mathbf{y})\|_2. \tag{33}$$

Then

$$\|\mathbf{U}^\top \mathbf{S}^\top \mathbf{S} \mathbf{U}(\tilde{\mathbf{y}} - \mathbf{y})\|_2 = \|\mathbf{U}^\top \mathbf{S}^\top (\mathbf{S}\mathbf{U}\tilde{\mathbf{y}} - \mathbf{S}\mathbf{b} + \mathbf{S}\mathbf{b} - \mathbf{S}\mathbf{U}\mathbf{y})\|_2 \tag{34}$$

$$= \|\mathbf{U}^\top \mathbf{S}^\top \mathbf{S} (\mathbf{U}\mathbf{y} - \mathbf{b})\|_2 \tag{35}$$

where the second equality follows because $\mathbf{S}\mathbf{U}\tilde{\mathbf{y}} - \mathbf{S}\mathbf{b}$ is orthogonal to any vector in the span of $\mathbf{S}\mathbf{U}$. By similar reasoning, notice that $\mathbf{U}^\top (\mathbf{U}\mathbf{y} - \mathbf{b}) = \mathbf{0}$. Then, as long as $m = O(\frac{n}{\delta\epsilon})$, we have

$$\|\mathbf{U}^\top \mathbf{S}^\top \mathbf{S} (\mathbf{U}\mathbf{y} - \mathbf{b})\|_2 \leq \frac{\sqrt{\epsilon}}{2\sqrt{n}} \|\mathbf{U}\|_F \|\mathbf{U}\mathbf{y} - \mathbf{b}\|_2 \tag{36}$$

with probability $1 - \delta/2$ by Lemma A.4. Since $\mathbf{U}$ has orthonormal columns, $\|\mathbf{U}\|_F^2 \leq n$. Then, combining inequalities yields

$$\|\mathbf{A}\tilde{\phi} - \mathbf{A}\phi\|_2^2 = \|\tilde{\mathbf{y}} - \mathbf{y}\|_2^2 \leq 2\|\mathbf{U}^\top \mathbf{S}^\top \mathbf{S} \mathbf{U}(\tilde{\mathbf{y}} - \mathbf{y})\|_2^2 \leq \epsilon \|\mathbf{U}\mathbf{y} - \mathbf{b}\|_2^2 = \epsilon \|\mathbf{A}\phi - \mathbf{b}\|_2^2 \tag{37}$$

with probability $1 - \delta$. $\qquad\square$

## B  PROOF OF APPROXIMATION COROLLARY

We will establish several properties specific to our problem and then use these properties to prove Corollary 4.1.

**Lemma B.1** (Properties of $\mathbf{A}$, $\phi$, and $\tilde{\phi}$). *The Shapley values $\phi$, the matrix $\mathbf{A}$, and the estimated Shapley values $\tilde{\phi}$ produced by Algorithm 1 satisfy*

$$\|\mathbf{A}(\tilde{\phi} - \phi)\|_2^2 = \frac{1}{n}\|\tilde{\phi} - \phi\|_2^2. \tag{38}$$

*and*

$$\|\mathbf{A}\phi\|_2^2 = \frac{1}{n}\left(\|\phi\|_2^2 - \frac{(v(\mathbf{1}) - v(\mathbf{0}))^2}{n}\right). \tag{39}$$

*Proof.* Even though $\mathbf{A}$ does not have $n$ singular values, we can exploit the special structure of $\mathbf{A}$, $\phi$, and $\tilde{\phi}$. Let $\mathbf{A} = \mathbf{U}\boldsymbol{\Sigma}\mathbf{V}^\top$ be the singular value decomposition of $\mathbf{A}$. By Lemma 3.3, we know $\boldsymbol{\Sigma} \in \mathbb{R}^{(n-1)\times(n-1)}$ is a diagonal matrix with $\frac{1}{\sqrt{n}}$ on the diagonal and $\mathbf{V} \in \mathbb{R}^{n\times(n-1)}$ has $n-1$ orthonormal columns that are all orthogonal to $\mathbf{1}$. Let $\mathbf{v}_1, \ldots, \mathbf{v}_{n-1} \in \mathbb{R}^n$ be these vectors. Then $\mathbf{A} = \sum_{i=1}^n \frac{1}{\sqrt{n}}\mathbf{u}_i\mathbf{v}_i^\top$ where $\mathbf{u}_1, \ldots, \mathbf{u}_{n-1} \in \mathbb{R}^{2^{n-1}}$ are the $n-1$ orthonormal columns in $\mathbf{U}$.

By Lemma 2.1 and 3.1, we can write

$$\phi = \sum_{i=1}^{n-1} \mathbf{v}_i s_i + \mathbf{1}\frac{v(\mathbf{1}) - v(\mathbf{0})}{n} \qquad \text{and} \qquad \tilde{\phi} = \sum_{i=1}^{n-1} \mathbf{v}_i \tilde{s}_i + \mathbf{1}\frac{v(\mathbf{1}) - v(\mathbf{0})}{n} \tag{40}$$

for some $s_i$ and $\tilde{s}_i$ where $i \in [n-1]$. Then $(\tilde{\phi} - \phi) = \sum_{i=1}^{n-1} \mathbf{v}_i(\tilde{s}_i - s_i)$ and

$$\mathbf{A}(\tilde{\phi} - \phi) = \sum_{i=1}^{n-1} \frac{1}{\sqrt{n}}\mathbf{u}_i\mathbf{v}_i^\top \sum_{j=1}^{n-1} \mathbf{v}_j(\tilde{s}_j - s_j) = \sum_{i=1}^{n-1} \frac{1}{\sqrt{n}}\mathbf{u}_i(\tilde{s}_i - s_i) \tag{41}$$

so $\|\mathbf{A}(\tilde{\phi} - \phi)\|_2^2 = \frac{1}{n}\|\tilde{\phi} - \phi\|_2^2$.

Similarly, we can write

$$\mathbf{A}\phi = \sum_{i=1}^{n-1} \frac{1}{\sqrt{n}}\mathbf{u}_i\mathbf{v}_i^\top \left(\sum_{j=1}^{n-1} \mathbf{v}_j s_j + \mathbf{1}\frac{v(\mathbf{1}) - v(\mathbf{0})}{n}\right) = \sum_{i=1}^{n-1} \frac{1}{\sqrt{n}}\mathbf{u}_i s_i \tag{42}$$

where the second equality follows because $\mathbf{v}_1, \ldots, \mathbf{v}_{n-1}$ are orthogonal to $\mathbf{1}$. Then $\|\mathbf{A}\phi\|_2^2 = \frac{1}{n}\left(\|\phi\|_2^2 - \frac{(v(\mathbf{1}) - v(\mathbf{0}))^2}{n}\right)$.

$\square$

**Corollary 4.1.** *Suppose $\tilde{\phi}$ satisfies $\|\mathbf{A}\tilde{\phi} - \mathbf{b}\|_2^2 \le (1+\epsilon)\|\mathbf{A}\phi - \mathbf{b}\|_2^2$. Let $\gamma = \|\mathbf{A}\phi - \mathbf{b}\|_2^2/\|\mathbf{A}\phi\|_2^2$. Then*

$$\|\phi - \tilde{\phi}\|_2^2 \le \epsilon\gamma\|\phi\|_2^2.$$

*Proof of Corollary 4.1.* We have

$$\|\mathbf{A}\tilde{\phi} - \mathbf{b}\|_2^2 = \|\mathbf{A}\tilde{\phi} - \mathbf{A}\phi + \mathbf{A}\phi - \mathbf{b}\|_2^2 = \|\mathbf{A}\tilde{\phi} - \mathbf{A}\phi\|_2^2 + \|\mathbf{A}\phi - \mathbf{b}\|_2^2 \tag{43}$$

where the second equality follows because $\mathbf{A}\phi - \mathbf{b}$ is orthogonal to any vector in the span of $\mathbf{A}$.

Then, by the assumption, we have

$$\|\mathbf{A}\tilde{\phi} - \mathbf{A}\phi\|_2^2 \le \epsilon\|\mathbf{A}\phi - \mathbf{b}\|_2^2. \tag{44}$$

By the definition of $\gamma = \frac{\|\mathbf{A}\phi - \mathbf{b}\|_2^2}{\|\mathbf{A}\phi\|_2^2}$ and Lemma B.1,

$$\epsilon\|\mathbf{A}\phi - \mathbf{b}\|_2^2 = \epsilon\gamma\|\mathbf{A}\phi\|_2^2 = \epsilon\gamma\frac{1}{n}\left(\|\phi\|_2^2 - \frac{(v(\mathbf{1}) - v(\mathbf{0}))^2}{n}\right). \tag{45}$$

Finally, by Lemma B.1 along with Equations 44 and 45, we have

$$\frac{1}{n}\|\tilde{\phi} - \phi\|_2^2 = \|\mathbf{A}(\tilde{\phi} - \phi)\|_2^2 \leq \epsilon\|\mathbf{A}\phi - \mathbf{b}\|_2^2 \leq \frac{1}{n}\epsilon\gamma\|\phi\|_2^2. \tag{46}$$

The corollary statement then follows after multiplying both sides by $n$. □

## C  COMPUTATIONAL HARDNESS

When the Shapley value problem is viewed as an optimization problem, we provide a constant factor approximation in polynomial time (see e.g., Theorem 1.1).

On the other hand, suppose we wanted to, e.g., ensure that $\frac{1}{C}\phi_i \leq \tilde{\phi}_i \leq C\phi_i$ for all $i$. We claim that obtaining this goal for any constant $C$ is NP-hard in many settings. In particular, to ask about computational hardness, we also need to specify the input to the algorithm: i.e., is $v$ given as a polynomial size circuit, a polynomial size formula, or as a black-box, unit cost oracle? In all of these settings the problem is NP-hard to approximate. In particular, consider the case when $v$ either evaluates to 0 for all sets, or evaluates to 0 for all sets except for a single set $S$. In the first case, the Shapley values will all equal 0, whereas in the second case, the Shapley values for indices in $S$ are non-zero. As such, to obtain a multiplicative approximation, we need to determine if there is some set $S$ for which $v$ does not evaluate to 0. When $v$ is a black-box, finding such a set clearly requires $\Omega(2^n)$ time (we can only enumerate all possible inputs). However, it is still hard when $v$ is given other forms. For example, if $v$ is a given as a circuit, then determining if there is an S for which $v(S) \neq 0$ is equivalent to the NP-complete circuit SAT problem.

This argument makes it clear why asking for a multiplicative approximation is not reasonable, and why instead we might care about an approximation in objective value. This is the case for many computational problems: for example, in the class of NP-hard optimization problems like set cover, we do not typically ask to approximate the optimal solution itself, but instead to find another solution with nearly the same objective value as the optimal solution (whether or not it is similar to that optimal solution or not).

## D  NOISY ACCESS TO THE SET FUNCTION

In this section, we explore how the regression-based estimators perform given noisy access to the set function $v$. This setting is particularly relevant for Shapley values in explainable AI, since the set function may be an expectation that is approximated in practice. Figure 4 shows plots performance as noise is added to the set functions: Instead of observing $v(S)$, the estimators observe $v(S) + \zeta$ where $\zeta \sim \mathcal{N}(0, \sigma^2)$ is normally distributed with standard deviation $\sigma$ (we set $\sigma = 0, 5 \times 10^{-3}, 1 \times 10^{-2}, 5 \times 10^{-2}, \ldots, 1$).

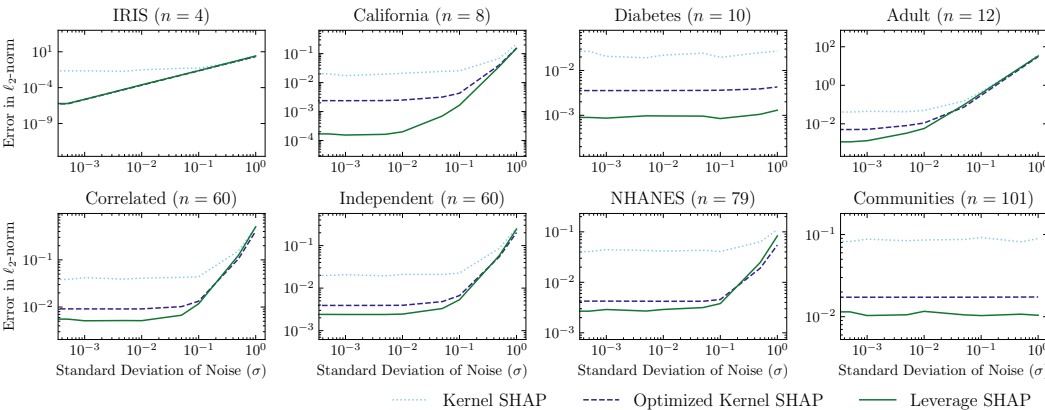

Figure 4: The $\ell_2$-norm error between the estimated and ground truth Shapley values as a function of noise in the set function. Leverage SHAP gives the best performance in all settings; the third quartile of the Leverage SHAP generally matches the median of the Optimized Kernel SHAP error.

# E  ERROR RELATIVE TO OPTIMIZED KERNEL SHAP

| Dataset | $5n$ | $10n$ | $20n$ | $40n$ | $80n$ | $160n$ |
|---|---|---|---|---|---|---|
| IRIS | 1.00 | 1.00 | 1.00 | 1.00 | 1.00 | 1.00 |
| California | 0.0923 | 0.0909 | 0.0463 | 1.00 | 1.00 | 1.00 |
| Diabetes | 0.314 | 0.247 | 0.278 | 0.203 | 0.110 | 1.00 |
| Adult | 0.351 | 0.482 | 0.536 | 0.318 | 0.279 | 0.276 |
| Correlated | 0.484 | 0.614 | 0.613 | 0.658 | 0.682 | 0.685 |
| Independent | 0.540 | 0.608 | 0.614 | 0.637 | 0.658 | 0.651 |
| NHANES | 0.572 | 0.702 | 0.757 | 0.709 | 0.658 | 0.741 |
| Communities | 0.514 | 0.668 | 0.647 | 0.681 | 0.661 | 0.722 |

Table 2: The table reports the average error ratio over 100 random runs between Leverage SHAP and Optimized Kernel SHAP. For example, on the *Correlated* dataset with $m = 5n$ samples, Leverage SHAP achieved estimates with $54\%$ of the error of Optimized Kernel SHAP. A ratio of 1 indicates that $m \geq 2^n$, and both algorithms recovered estimates accurate to machine precision. On average (excluding entries where the ratio is 1), Leverage SHAP estimates have $50.2\%$ of the error of Optimized Kernel SHAP. In other words, Leverage SHAP achieves roughly a $50\%$ reduction in error compared to Optimized Kernel SHAP.

Table 2 presents the average error ratios between Leverage SHAP and Optimized Kernel SHAP across various datasets and sample sizes, computed over 100 independent random runs. Each entry reports the ratio of the mean absolute error of Leverage SHAP to that of Optimized Kernel SHAP. For example, on the *Correlated* dataset with $m = 5n$ samples, Leverage SHAP achieved an error that was $54\%$ of the error of Optimized Kernel SHAP. A reported ratio of 1 corresponds to cases where $m \geq 2^n$, in which both methods recovered estimates accurate to machine precision. Averaging over all non-unit entries, Leverage SHAP achieved $50.2\%$ of the error of Optimized Kernel SHAP, corresponding to an approximate $50\%$ reduction in error

## F    EXPLORING $\gamma$

The problem-specific term $\gamma = \|\mathbf{A}\phi - \mathbf{b}\|_2^2/\|\mathbf{b}\|_2^2$ appears in the $\ell_2$-norm guarantee of Corollary 4.1. In this section, we explore the distribution of $\gamma$, and the performance by $\gamma$. Because exactly computing $\gamma$ requires evaluating $\mathbf{b}$, we focus on datasets where $n \leq 16$.

Table 3 shows summary statistics of $\gamma$ over 100 runs (the randomness comes from the different models trained on the data). On every dataset, the third quartile of $\gamma$ is less than 2.

Table 3: Summary statistic of $\gamma$ over 100 runs.

| Dataset | $n$ | 1st Quartile | 2nd Quartile | 3rd Quartile |
|---------|-----|--------------|--------------|--------------|
| IRIS | 4 | 0.000234 | 0.266 | 1.03 |
| California | 8 | 0.158 | 0.298 | 0.449 |
| Diabetes | 10 | 0.174 | 0.321 | 0.513 |
| Adult | 12 | 0.180 | 0.395 | 0.703 |

In Figure 5, we explore experimentally whether $\gamma$ is an artifact of our analysis or a necessary parameter. We perform the experiment by explicitly building the matrix $\mathbf{A}$ (only possible for small $n$) and then choosing $\mathbf{b}$ as a linear combination of a vector in the column span of $\mathbf{A}$ and a vector not in the column span of $\mathbf{A}$. Then, we back out the corresponding set function $v$ using the definition of $\mathbf{b}$. We induce different values of $\gamma$ as we vary how much of $\mathbf{b}$ is a vector in the column span of $\mathbf{A}$. Because all three of the algorithms we consider perform worse as $\gamma$ increases, the experiment suggests that $\gamma$ is not an artifact of our analysis. The same trend appears for the ablated estimators, as shown in Figure 6. As our analysis suggests, $\gamma$ does not appear to be a relevant factor for the objective error, as shown in Figure 7.

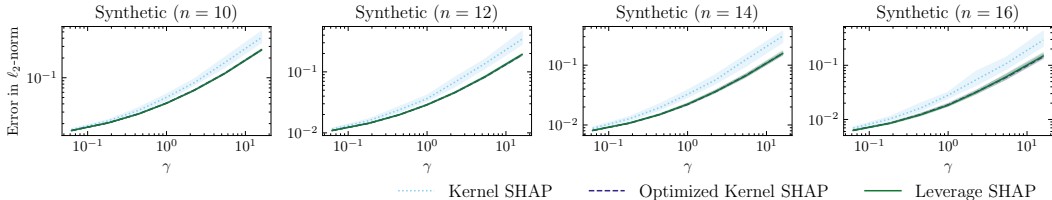

Figure 5: The $\ell_2$-norm error as a function of $\gamma$, the problem-specific parameter in Corollary 4.1. The lines indicate the median and the shaded regions encompass the first and third quartile over 100 runs. All three of the main algorithms we consider have higher $\ell_2$-norm error as $\gamma$ grows, suggesting that $\gamma$ is not an artifact of our analysis.

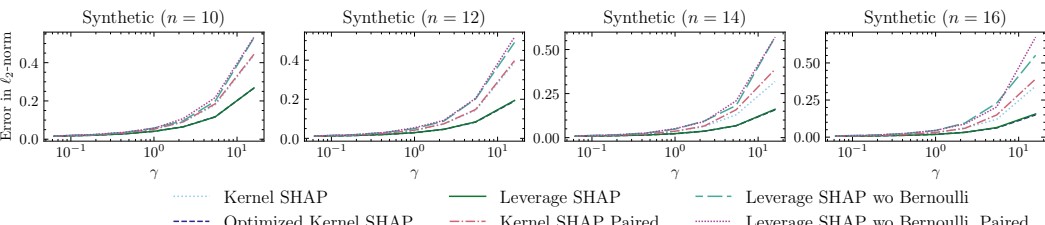

Figure 6: The $\ell_2$-norm error as a function of $\gamma$, the problem-specific parameter in Corollary 4.1. The lines indicate the mean over 100 runs. All algorithms we consider have higher $\ell_2$-norm error as $\gamma$ grows, suggesting that $\gamma$ is not an artifact of our analysis.

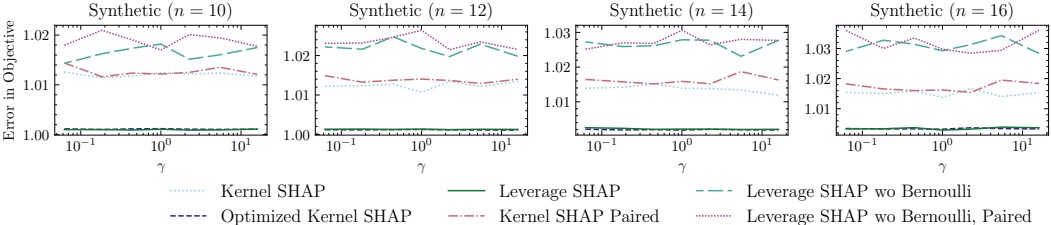

Figure 7: The linear objective error as a function of $\gamma$, the problem-specific parameter in Corollary 4.1. The lines indicate the mean over 100 runs. All algorithms we consider have no dependence on $\gamma$ grows, suggesting that $\gamma$ is not a relevant parameter for the linear objective error. This finding aligns with Theorem 1.1, which has no dependence on $\gamma$.

# G  ADDITIONAL EXPERIMENTS WITH $\ell_2$-NORM ERROR

Table 4 evaluates additional estimators. Leverage SHAP continues to give the best performance, notably outperforming Permutation SHAP.

| | IRIS | California | Diabetes | Adult | Correlated | Independent | NHANES | Communities |
|---|---|---|---|---|---|---|---|---|
| **Kernel SHAP** | | | | | | | | |
| Mean | 0.026 | 0.0266 | 0.0553 | 0.0673 | 0.0465 | 0.0264 | 0.0604 | 0.12 |
| 1st Quartile | 1.61e-05 | 0.00829 | 0.0116 | 0.0182 | 0.0244 | 0.0134 | 0.0202 | 0.0563 |
| 2nd Quartile | 0.000898 | 0.0236 | 0.0229 | 0.0345 | 0.0404 | 0.0217 | 0.0388 | 0.089 |
| 3rd Quartile | 0.0328 | 0.0424 | 0.0524 | 0.0936 | 0.056 | 0.0303 | 0.0843 | 0.149 |
| **Optimized Kernel SHAP** | | | | | | | | |
| Mean | 4.84e-09 | 0.00342 | 0.0093 | 0.00989 | 0.0117 | 0.00474 | 0.00758 | 0.0233 |
| 1st Quartile | 1.66e-13 | 0.000802 | 0.00161 | 0.00187 | 0.00499 | 0.00194 | 0.00156 | 0.00962 |
| 2nd Quartile | 2.17e-13 | 0.00238 | 0.00356 | 0.00489 | 0.00916 | 0.00391 | 0.00425 | 0.0173 |
| 3rd Quartile | 2.69e-10 | 0.00489 | 0.00868 | 0.0122 | 0.015 | 0.00695 | 0.00871 | 0.0325 |
| **Leverage SHAP** | | | | | | | | |
| Mean | 4.84e-09 | 0.000311 | 0.0023 | 0.00477 | 0.00716 | 0.00288 | 0.00532 | 0.0156 |
| 1st Quartile | 1.66e-13 | 4.47e-05 | 0.000215 | 0.000477 | 0.00289 | 0.000843 | 0.000995 | 0.0062 |
| 2nd Quartile | 2.17e-13 | 0.000133 | 0.000969 | 0.00124 | 0.00528 | 0.00257 | 0.00288 | 0.0104 |
| 3rd Quartile | 2.69e-10 | 0.000366 | 0.00241 | 0.00354 | 0.00891 | 0.00417 | 0.00554 | 0.0225 |
| **Kernel SHAP Paired** | | | | | | | | |
| Mean | 0.00343 | 0.00081 | 0.00437 | 0.00607 | 0.0107 | 0.00453 | 0.00754 | 0.0249 |
| 1st Quartile | 3.07e-10 | 9.92e-05 | 0.000444 | 0.000676 | 0.00395 | 0.00143 | 0.00156 | 0.0108 |
| 2nd Quartile | 1.28e-07 | 0.000361 | 0.00203 | 0.00157 | 0.00827 | 0.00344 | 0.00415 | 0.016 |
| 3rd Quartile | 0.000189 | 0.000971 | 0.00565 | 0.00482 | 0.0141 | 0.00658 | 0.00857 | 0.034 |
| **Leverage SHAP (Unpaired)** | | | | | | | | |
| Mean | 0.0332 | 0.0237 | 0.049 | 0.0531 | 0.0308 | 0.0163 | 0.0357 | 0.067 |
| 1st Quartile | 9.5e-06 | 0.00834 | 0.00947 | 0.0144 | 0.0171 | 0.00775 | 0.00969 | 0.0322 |
| 2nd Quartile | 0.000951 | 0.0183 | 0.0184 | 0.0303 | 0.0245 | 0.013 | 0.024 | 0.0495 |
| 3rd Quartile | 0.0277 | 0.0328 | 0.035 | 0.067 | 0.0356 | 0.02 | 0.0479 | 0.0774 |
| **Permutation SHAP** | | | | | | | | |
| Mean | 0.00511 | 0.00145 | 0.00875 | 0.0118 | 0.015 | 0.00517 | 0.00868 | 0.0312 |
| 1st Quartile | 1.29e-09 | 0.000174 | 0.000785 | 0.000708 | 0.00403 | 0.00109 | 0.00127 | 0.0105 |
| 2nd Quartile | 2.35e-07 | 0.000793 | 0.00361 | 0.00222 | 0.0098 | 0.00338 | 0.00355 | 0.0188 |
| 3rd Quartile | 0.000659 | 0.00196 | 0.0102 | 0.0079 | 0.0199 | 0.00847 | 0.00789 | 0.045 |

Table 4: Normalized $\ell_2$-norm by dataset and estimator, reported in summary statistics from 100 random runs.

Figures 8, and 9 show additional experiments where the error is measured with the $\ell_2$-norm error. We find that Leverage SHAP gives the best performance.

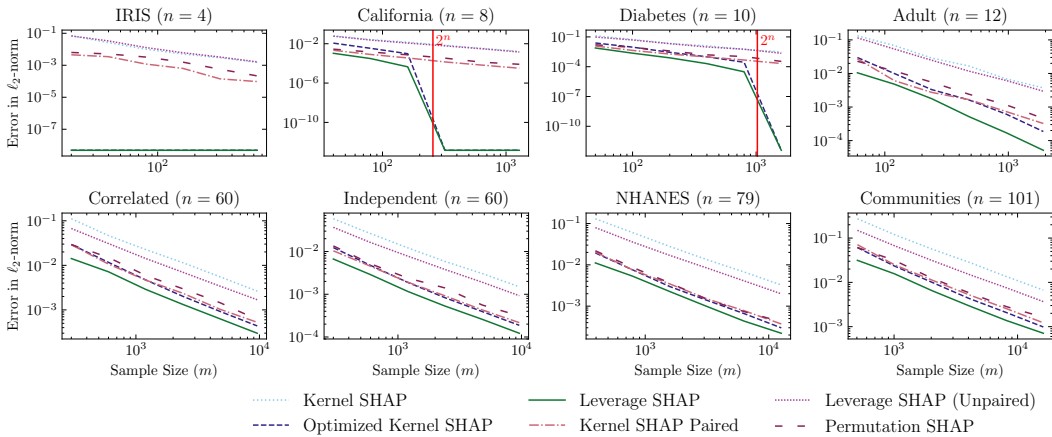

Figure 8: The $\ell_2$-norm error between the estimated Shapley values and ground truth Shapley values as a function of sample size. The lines report the mean error over 100 runs. Leverage SHAP reliably gives the best performance, exceeding second-best Optimized Kernel SHAP for small $n$ and Leverage SHAP w/o Bernoulli Sampling for large $n$.

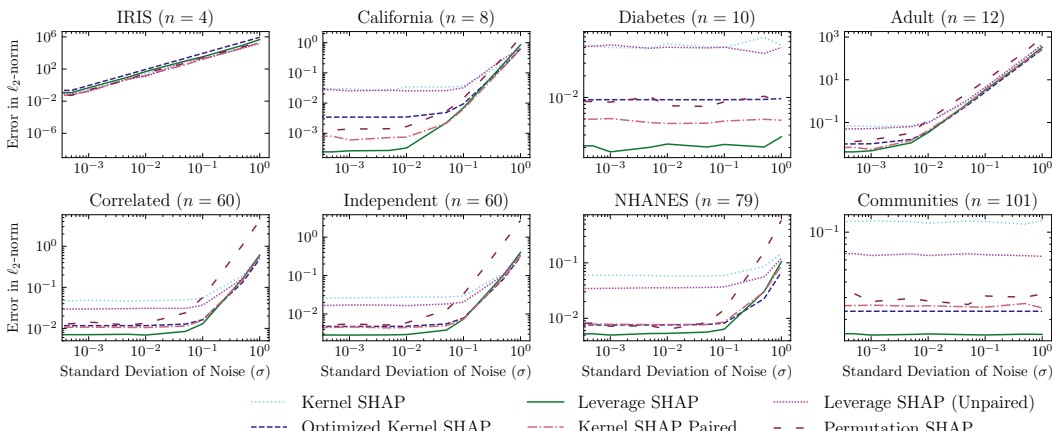

Figure 9: The $\ell_2$-norm error between the estimated and ground truth Shapley values as a function of noise in the set function. Leverage SHAP gives the best performance in almost all settings, with the exception of NHANES where Leverage SHAP with replacement (without Bernoulli) gives slightly better performance.

## H  ABLATION EXPERIMENTS WITH OBJECTIVE ERROR

Figures 10 and 11 show ablation experiments where the error is measured with the objective value naturally suggested by the regression formulation. We find that Leverage SHAP, or its ablated version without Bernoulli sampling, give the best performance.

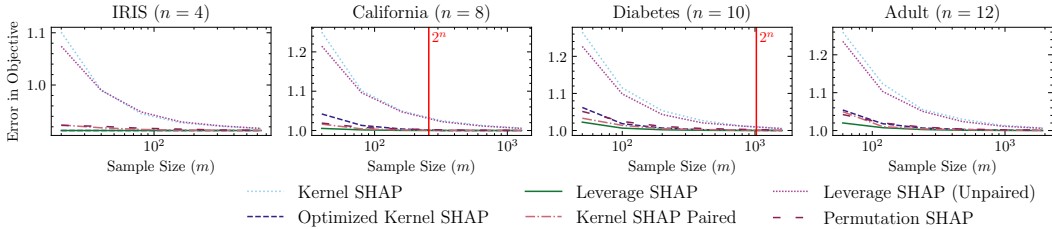

Figure 10: Linear objective error by sample size. The lines indicate the mean error over 100 runs. Leverage SHAP quickly achieves the lowest error in all settings. (For small $n$, numerical instability can lead to error below 1.)

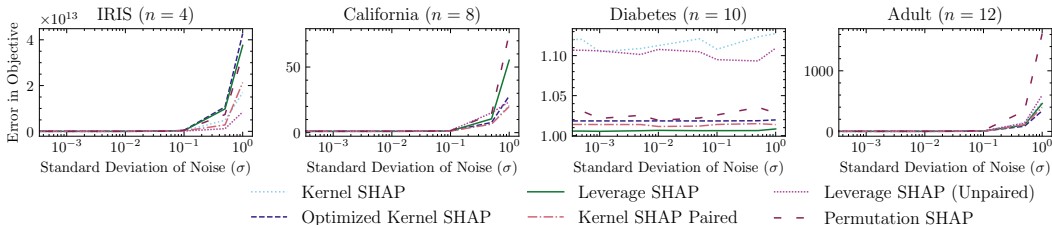

Figure 11: Linear objective error by standard deviation of noise. The noise grows large as $\sigma$ approaches one for all algorithms. The plots indicate no clear winner across all four datasets.

## I   CONSTRAINED AND UNCONSTRAINED EQUIVALENCE

**Lemma 2.1** (Equivalence (Lundberg & Lee, 2017; Charnes et al., 1988)).

$$\phi = \underset{\mathbf{x}:\langle\mathbf{x},\mathbf{1}\rangle=v(\mathbf{1})-v(\mathbf{0})}{\arg\min} \|\mathbf{Z}\mathbf{x}-\mathbf{y}\|_2^2 \tag{6}$$

$$= \underset{\mathbf{x}:\langle\mathbf{x},\mathbf{1}\rangle=v(\mathbf{1})-v(\mathbf{0})}{\arg\min} \sum_{\mathbf{z}:0<\|\mathbf{z}\|_1<n} w(\|\mathbf{z}\|_1)\cdot(\langle\mathbf{z},\mathbf{x}\rangle-(v(\mathbf{z})-v(\mathbf{0})))^2. \tag{7}$$

We provide a cleaner, direct proof using the unconstrained problem and the exact characterization of $\mathbf{Z}^\top\mathbf{Z}$.

*Proof of Lemma 2.1.* We will show that

$$\phi = \underset{\mathbf{x}}{\arg\min}\|\mathbf{A}\mathbf{x}-\mathbf{b}\|_2^2 + \mathbf{1}\frac{v(\mathbf{1})-v(\mathbf{0})}{n} \tag{47}$$

where $\mathbf{A}=\mathbf{Z}\mathbf{P}$ and $\mathbf{b}=\mathbf{y}-\mathbf{Z}\mathbf{1}\frac{v(\mathbf{1})-v(\mathbf{0})}{n}$. Then Lemma 2.1 follows from Lemma 3.1.

We know

$$\underset{\mathbf{x}}{\arg\min}\|\mathbf{A}\mathbf{x}-\mathbf{b}\|_2^2 = (\mathbf{A}^\top\mathbf{A})^+\mathbf{A}^\top\mathbf{b}. \tag{48}$$

By Lemma 3.3, we have $(\mathbf{A}^\top\mathbf{A})^+ = (\mathbf{P}^\top\mathbf{Z}^\top\mathbf{Z}\mathbf{P})^+ = n\mathbf{I}-\mathbf{1}\mathbf{1}^\top = n\mathbf{P}$. Then

$$(\mathbf{A}^\top\mathbf{A})^+\mathbf{A}^\top = n\mathbf{P}\mathbf{P}^\top\mathbf{Z}^\top = n\mathbf{P}\mathbf{Z}^\top. \tag{49}$$

Let $\mathbf{z}\in\{0,1\}^n$ with $0<\|\mathbf{z}\|_1<n$. It follows that

$$(\mathbf{A}^\top\mathbf{A})^+\mathbf{A}^\top\mathbf{b} = n\mathbf{P}\left(\mathbf{Z}^\top\begin{bmatrix}\vdots\\\sqrt{w(\|\mathbf{z}\|_1)}\,(v(\mathbf{z})-v(\mathbf{0}))\\\vdots\end{bmatrix} - \mathbf{Z}^\top\begin{bmatrix}\vdots\\\sqrt{w(\|\mathbf{z}\|_1)}\|\mathbf{z}\|_1\frac{v(\mathbf{1})-v(\mathbf{0})}{n}\\\vdots\end{bmatrix}\right).$$

By symmetry, the last term is a scaling of the all ones vector so it gets projected off by $\mathbf{P}$. Recall $w(s) = \frac{(s-1)!(n-s-1)!}{n!}$. The weights are carefully designed so that $w(s)s = w(s+1)(n-s-1)$. Then, for $i \in [n]$, we have the $i$th entry of $(\mathbf{A}^\top \mathbf{A})^+ \mathbf{A}^\top \mathbf{b}$ given by

$$\sum_{\mathbf{z}:0<\|\mathbf{z}\|_1<n} w(\|\mathbf{z}\|_1)(nz_i - \|\mathbf{z}\|_1)\left(v(\mathbf{z}) - v(\mathbf{0})\right)$$

$$= \frac{1}{n}\sum_{S:\{i\}\subseteq S\subset[n]} \binom{n-1}{s}^{-1} v(S) - \frac{1}{n}\sum_{S:\emptyset\subset S\subseteq[n]\setminus\{i\}} \binom{n-1}{s}^{-1} v(S) \tag{50}$$

$$+ \frac{v([n]) - v(\emptyset)}{n} - \frac{v([n]) - v(\emptyset)}{n} \tag{51}$$

$$= \frac{1}{n}\sum_{S:\{i\}\subseteq S\subseteq[n]} \binom{n-1}{s}^{-1} v(S) - \frac{1}{n}\sum_{S:\emptyset\subseteq S\subseteq[n]\setminus\{i\}} \binom{n-1}{s}^{-1} v(S) \tag{52}$$

$$- \frac{v([n]) - v(\emptyset)}{n} \tag{53}$$

$$= \phi_i - \frac{v([n]) - v(\emptyset)}{n} \tag{54}$$

where the first equality follows by converting from binary vectors to sets. Then Equation 47 follows.

$\square$

## J EFFICIENTLY SAMPLING WITHOUT REPLACEMENT

Since there are an exponential number of rows, the naive method of considering each row independently will not work. Our approach is as follows: For each set size $s \in [\lfloor n/2 \rfloor]$, we sample the number of samples that will be chosen. This is distributed as a Binomial distribution with $\binom{n}{s}$ trials and probability $\min(1, 2c\binom{n}{s}^{-1})$. Then, because the probabilities are the same for all subsets of size $s$, we exploit this symmetry and sample random indices uniformly from $[\binom{n}{s}]$ without replacement. We index all subsets of size $s$ and then construct the subsets corresponding to the randomly sampled indices without enumerating all subsets. The Bernoulli sampling code appears in Algorithm 2 and the subset construction code appears in Algorithm 3, both deferred to Appendix J.

Like before, there is one special case when $n$ is even and there is a middle set size $s = \lfloor n/2 \rfloor$. Here, the paired samples $\mathbf{z}$ and $\bar{\mathbf{z}}$ have the same size $s$ so we need to partition the set of all subsets of size $s$ so we do not risk sampling the same pair twice. We accomplish this by sampling $\mathbf{z}$ from all subsets of $n-1$ items with size $s-1$ then appending $\mathbf{z}_n = 1$ and computing the complement $\bar{\mathbf{z}}$.

**Lemma J.1** (Time Complexity). *Let $m'$ be the number of samples used in Algorithm 1 and define $T_{m'}$ as the time complexity of evaluating $v$ in parallel on $m'$ inputs. Algorithm 1 runs in time $O(m'n^2 + T_{m'})$.*

For most settings, we expect $T_{m'}$ to dominate the time complexity. For example, consider a shallow fully connected neural network with inner dimension just $n$. Even evaluating the forward pass on $m'$ inputs for a single layer of this network takes time $O(m'n^2)$.

*Proof of Lemma J.1.* `BernoulliSample` takes $O(mn^2)$ time: For each pair of samples, we call `Combo` at most twice which takes time at most $O(n^2)$ since there are only two loops, each over at most $n$ items. For simplicity of presentation, the algorithm calls $v$ three times: once with input $\mathbf{1}$, once with input $\mathbf{0}$, and once with $m'-2$ inputs. However, by concatenating the inputs, the algorithm can be easily modified to call $v$ once on $m'$ inputs for a time complexity of $T_{m'}$. Computing the optimal subsampled solution $\tilde{\phi}^\perp$ takes $O(n^3)$ time to compute a factorization of a $n \times n$ matrix and $O(m'n^2)$ time to compute the $n \times n$ matrix. Since $m' \geq n$, the time complexity is $O(m'n^2)$. $\square$

---

**Algorithm 2** `BernoulliSample`$(n, c)$: Efficient Bernoulli Sampling

---

**Input:** $n$: number of players, $c$: probability scaling
**Output:** $\mathcal{Z}'$: random sample of $\mathcal{Z}$ that independently contains pair $(\mathbf{z}, \bar{\mathbf{z}})$ w.p. $\min(1, 2c \cdot \ell_z)$
1: $\mathcal{Z} \leftarrow \emptyset$
2: **for** set size $s \in \{1, \ldots, \lfloor \frac{n}{2} \rfloor\}$ **do**
3:      isMiddle $\leftarrow (2|n) \wedge (s = \lfloor \frac{n}{2} \rfloor)$      ▷ Special handling of middle set size for paired samples
4:      $m_s \sim \text{Binomial}(\binom{n}{s}, \min(1, 2c \cdot \binom{n}{s}^{-1}))$
5:      $m_s \leftarrow \lfloor m_s/2 \rfloor$ if isMiddle
6:      randomIndices $\leftarrow m_s$ uniform random integers drawn without replacement from $[\binom{n}{s}]$
7:      **for** $i \in$ randomIndices **do**
8:          $\mathbf{z} \leftarrow \text{Combo}(n, s, i)$      ▷ $i$th combination of $n$ items with size $s$ (Algorithm 3)
9:          **if** isMiddle **then**      ▷ Partition the middle size by setting $\mathbf{z}_n = 1$
10:              $\mathbf{z} \leftarrow \text{Combo}(n - 1, s - 1, i)$      ▷ $i$th combination of $n - 1$ items with size $s - 1$
11:              Append 0 to $\mathbf{z}$
12:          **end if**
13:          Add $(\mathbf{z}, \bar{\mathbf{z}})$ to $\mathcal{Z}'$
14:      **end for**
15: **end for**
16: **return** $\mathcal{Z}'$

---

**Algorithm 3** `Combo`$(n, s, i)$: Compute the $i$th Combination in Lexicographic Order

---

1: **Input:** $n$: total number of elements, $s$: subset size, $i$: index of subset
2: **Output:** $\mathbf{z}$: the $i$th combination (lexicographically) in binary form
3: $\mathbf{z} \leftarrow \mathbf{0}_n, k \leftarrow s, \text{start} \leftarrow 1$
4: **for** idx $\in \{1, \ldots, s\}$ **do**
5:      **for** $j \in \{\text{start}, \ldots, n\}$ **do**
6:          count $\leftarrow \binom{n-s-1}{k-1}$      ▷ Combinations possible with remaining elements if $j$ is added
7:          **if** $i <$ count **then**
8:              $z_j \leftarrow 1$      ▷ Add $j$
9:              $k \leftarrow k - 1$      ▷ Decrease the number of elements left to choose
10:              start $\leftarrow s + 1$      ▷ Update starting index to ensure lexicographic order
11:              **break**
12:          **else**
13:              $i \leftarrow i -$ count      ▷ Skip this batch of combinations
14:          **end if**
15:      **end for**
16: **end for**
17: **return** $\mathbf{z}$

---

## K    CONSTRAINED REGRESSION

**Lemma 3.1** (Constrained to Unconstrained). *Let $\mathbf{P}$ be the projection matrix $\mathbf{I} - \frac{1}{n}\mathbf{1}\mathbf{1}^\top$. Define* $\mathbf{A} = \mathbf{ZP}$ *and* $\mathbf{b} = \mathbf{y} - \mathbf{Z}\mathbf{1}\frac{v(\mathbf{1})-v(\mathbf{0})}{n}$. *Then*

$$\underset{\mathbf{x}:\langle\mathbf{x},\mathbf{1}\rangle=v(\mathbf{1})-v(\mathbf{0})}{\arg\min}\|\mathbf{Zx}-\mathbf{y}\|_2^2 = \arg\min_{\mathbf{x}}\|\mathbf{Ax}-\mathbf{b}\|_2^2 + \mathbf{1}\frac{v(\mathbf{1})-v(\mathbf{0})}{n}. \tag{8}$$

*Further, we have that* $\min_{\mathbf{x}:\langle\mathbf{x},\mathbf{1}\rangle=v(\mathbf{1})-v(\mathbf{0})}\|\mathbf{Zx}-\mathbf{y}\|_2^2 = \min_{\mathbf{x}}\|\mathbf{Ax}-\mathbf{b}\|_2^2$.

*Proof of Lemma 3.1.* We will decompose $\mathbf{x}$ into one vector that is orthogonal to $\mathbf{1}$ and another vector that is a scaling of $\mathbf{1}$. That is, $\mathbf{x} = \mathbf{x}' + c\mathbf{1}$ where $\langle\mathbf{x}',\mathbf{1}\rangle = 0$ for some $c$. In order to satisfy the constraint, it follows that $c = \frac{v(\mathbf{1})-v(\mathbf{0})}{n}$. Then

$$\underset{\mathbf{x}:\langle\mathbf{x},\mathbf{1}\rangle=v(\mathbf{1})-v(\mathbf{0})}{\arg\min}\|\mathbf{Zx}-\mathbf{y}\|_2^2 = \underset{\mathbf{x}'+c\mathbf{1}:\langle\mathbf{x}',\mathbf{1}\rangle=0}{\arg\min}\|\mathbf{Z}(\mathbf{x}'+c\mathbf{1})-\mathbf{y}\|_2^2 \tag{55}$$

$$= \underset{\mathbf{x}':\langle\mathbf{x}',\mathbf{1}\rangle=0}{\arg\min}\|\mathbf{Zx}'-(\mathbf{y}-c\mathbf{Z1})\|_2^2 + c\mathbf{1} \tag{56}$$

$$= \arg\min_{\mathbf{x}'}\|\mathbf{ZPx}'-(\mathbf{y}-c\mathbf{Z1})\|_2^2 + c\mathbf{1} \tag{57}$$

We used $\mathbf{P}$ to project off any component in the direction of $\mathbf{1}$ and thereby remove the constraint. Plugging in the value of $c$ with the definitions of $\mathbf{A}$ and $\mathbf{b}$ yields the first equation. The second equation follows by a similar argument. $\square$

**Traditional Solution Used in Kernel SHAP**    Consider the problem

$$\mathbf{x}^* = \underset{\mathbf{x}:\langle\mathbf{x},\mathbf{1}\rangle=v(\mathbf{1})-v(\mathbf{0})}{\arg\min}\|\mathbf{Zx}-\mathbf{y}\|_2^2. \tag{58}$$

For Leverage SHAP, we reformulate the problem to an unconstrained regression problem with Lemma 3.1 and solve it using standard least squares.

In the standard Kernel SHAP implementation, the constrained problem is solved directly using the method of Lagrange multipliers. We put the analysis in here for completeness.

The Lagrangian function with respect to $\mathbf{x}$ and multiplier $\lambda \in \mathbb{R}$ is

$$\mathcal{L}(\mathbf{x},\lambda) = \|\mathbf{Zx}-\mathbf{y}\|_2^2 + \lambda(\langle\mathbf{x},\mathbf{1}\rangle - v(\mathbf{1}) + v(\mathbf{0})) \tag{59}$$

At the optimal solution $\mathbf{x}^*$ and $\lambda^*$, the gradients with respect to $\mathbf{x}$ and $\lambda$ are 0. This observation implies the following two equations:

$$\nabla_{\mathbf{x}}\mathcal{L}(\mathbf{x}^*,\lambda^*) = 2\mathbf{Z}^\top(\mathbf{Zx}^*-\mathbf{y}) + \lambda^*\mathbf{1} = 0 \quad\Longrightarrow\quad \mathbf{x}^* = (\mathbf{Z}^\top\mathbf{Z})^{-1}\left(\mathbf{Z}^\top\mathbf{y} - \frac{\lambda^*}{2}\mathbf{1}\right) \tag{60}$$

$$\nabla_{\lambda}\mathcal{L}(\mathbf{x}^*,\lambda^*) = \langle\mathbf{x}^*,\mathbf{1}\rangle - v(\mathbf{1}) + v(\mathbf{0}) = 0. \tag{61}$$

Together, Equations 60 and 61 tell us that

$$\langle\mathbf{1},\mathbf{x}^*\rangle = v(\mathbf{1}) - v(\mathbf{0}) = \mathbf{1}^\top(\mathbf{Z}^\top\mathbf{Z})^{-1}\left(\mathbf{Z}^\top\mathbf{y} - \frac{\lambda^*}{2}\mathbf{1}\right) \tag{62}$$

Plugging back into Equation 60, we have that

$$\mathbf{x}^* = (\mathbf{Z}^\top\mathbf{Z})^{-1}\left(\mathbf{Z}^\top\mathbf{y} - \frac{\mathbf{1}^\top(\mathbf{Z}^\top\mathbf{Z})^{-1}\mathbf{Z}^\top\mathbf{y} - v(\mathbf{1}) + v(\mathbf{0})}{\mathbf{1}^\top(\mathbf{Z}^\top\mathbf{Z})^{-1}\mathbf{1}}\mathbf{1}\right). \tag{63}$$

