# OpenReview forum: "Provably Accurate Shapley Value Estimation via Leverage Score Sampling"
_ICLR.cc/2025/Conference — ICLR 2025 Spotlight_

### Official Review · Reviewer_m993 · 2024-11-02

**Soundness:** 4
**Presentation:** 4
**Contribution:** 4
**Rating:** 8
**Confidence:** 3

**Summary:**

This paper modifies the Kernel SHAP algorithm, which approximates the Shapley value and model-agnostically quantifies the role of each feature in a model prediction. The Kernel SHAP did not enjoy theoretical convergence guarantees, and the authors propose a modification of this algorithm that offers a theoretical convergence guarantee and similar numerical performance.

**Strengths:**

This paper is very well written, introduces the context of their work beautifully, and provides both a theoretical and practical contribution to the field.

**Weaknesses:**

A weakness is that it might feel niche, but as a non-specialist in interpretable AI, I cannot judge the importance of the Shapley values. If this information is important, then the author's contribution is quite important because it removes some level of heuristic thanks to their theoretical contribution.

**Questions:**

N.A.

---

> ### Author Response · Authors · 2024-11-19
>
> Dear Reviewer m993,
>
> Thank you for your time and feedback!
>
> We would like to add some context to the importance of approximating Shapley values. In the explainable AI literature, Shapley values are the de facto method for providing explanations for machine learning predictions on tabular data. In particular, the original SHAP paper by Lundberg and Lee titled “A Unified Approach to Interpreting Model Predictions” has more than 25,000 citations. Further, the corresponding SHAP library (of which Kernel SHAP is probably the most widely utilized algorithm) has been used by more than 20,000 repositories, as reported by GitHub. As such, we believe Leverage SHAP has the potential for significant impact because of its superior empirical performance and its strong, non-asymptotic theoretical guarantees.

---

### Official Review · Reviewer_7UDD · 2024-11-06

**Soundness:** 3
**Presentation:** 2
**Contribution:** 2
**Rating:** 6
**Confidence:** 2

**Summary:**

The paper develops a computationally efficient method for approximating Shapely values, which are useful in interpretable machine learning. The proposed method provides a modification of the well-known Kernel SHAP method, with additional non-asymptotic theoretical convergence guarantees and better empirical performance. The authors use a modified sampling without replacement approach to optimize their method and report experiments with ablation studies for the optimizations.

**Strengths:**

- Estimating Shapley scores accurately and efficiently is an important problem in explainable machine learning. The paper provides a theoretically principled approach for this problem.
- The approach seems to outperform Kernel SHAP and optimized Kernel SHAP baselines in the experiments.

**Weaknesses:**

- The main theoretical result (Theorem 1.1) is somewhat unsatisfactory as it does not directly compare the true and estimated Shapely values. The authors address this via Corollary 4.1, but it has a non-intuitive $\gamma$ term which is can be large and makes the approximation guarantees weaker. Are there conditions under which $\gamma$ is guaranteed to be small? This would better help understand the limitations of current theoretical results.
- The experiments could include more baselines like (Jethani et al., 2021), (Mitchell et al., 2022b), and (Yang & Salman, 2019) for a more comprehensive comparison with the state-of-the-art.
- The technical novelty in proving the new theoretical results also appears to be limited. The main result seems similar to the active learning guarantee Theorem 1.1 of Shimizu et al. ICLR 2024, and it is not clear what additional technical insights are needed to develop the current result. A discussion of novel technical insights needed to develop the current result would be helpful.

Typos:
- Line 155, Line192 finte, Line 255 contrained
- References are incorrectly bracketed

**Questions:**

- Which algorithm in the literature does the Optimized Kernel SHAP in the experiments correspond to?
- Do the theoretical guarantees for leverage SHAP continue to hold when optimizations like paired sampling, and sampling without replacement are not applied? Is there a difference in the guarantees with and without the optimizations?
- In Table 2, Leverage SHAP w/o Bernoulli sampling seems to outperform in some cases. Is there a way to understand this? An analysis or discussion around this would be very useful to better understand the method's behavior.

---

> ### Author Response · Authors · 2024-11-19
>
> Dear Reviewer 7UDD,
>
> Thank you for your time and feedback! We address your concerns here and your questions in a comment below.
>
> > Corollary 4.1 [...] has a non-intuitive $\gamma$ term which is can be large and makes the approximation guarantees weaker.
>
> We agree that the dependence on $\gamma$ is not ideal, although dependence on this quantity seems inherent to all regression-based algorithms, including Kernel SHAP. This is illustrated in Figures 5 and 6. As discussed in our response to Reviewer LX1d, it would be interesting to consider lower bounds for Shapley value estimation in future work: it may be that the dependence on $\gamma$ is unavoidable for *any* method that, like our method, only uses a near-linear number of value function evaluations.
>
> > Are there conditions under which $\gamma$ is guaranteed to be small?
>
> This is an interesting question. $\gamma$ is known to equal $0$ for linear value functions, which covers e.g., the case when Shapley values are used for feature attribution in linear machine learning models (when the baseline is the all 0 vector). We are not sure if there are other more general conditions under which $\gamma$ can be bounded, but it is a nice question for future work.
>
> We do note that gamma is often quite small in experiments. Since computing $\gamma$ requires writing down the entire (exponentially) large regression problem, we can only compute $\gamma$ for datasets with small $n$. For each of these datasets, we train a model 100 times (from different random initializations) which induces a different vector $b$. Then we compute $\gamma = \| \| A \phi - b\|\|_2^2 / \|\| A \phi \|\|_2^2$. For all datasets, we found that the 75th percentile of $\gamma$ was less than 2. Please see a comment below for all the summary statistics.
>
> > The experiments could include more baselines like…
>
> Thank you for these suggestions. We address each paper below.
>
> > (Jethani et al., 2021)
>
> The Fast SHAP algorithm (Jethani et al., 2021) is complementary to our setting because the algorithm is trained (using many many value function evaluations) to predict Shapley values for multiple explicands and baselines. In contrast, the algorithms we consider, like Kernel SHAP and our Leverage SHAP method, estimate the Shapley values for a single value function $v$ corresponding to a single explicand and baseline pair without any pretraining. Which algorithm to use depends a lot on the setting and if the expensive  training cost of Fast SHAP can be amortized against enough Shapley value computations. We will add further discussion of this point to the next version of the paper.
>
> > (Mitchell et al., 2022b)
>
> We have added a comparison to the permutation algorithm (Strumbelj & Kononenko, 2010 and Mitchell et al., 2022), which we present in a comment below. We find that Permutation SHAP sometimes outperforms Optimized Kernel SHAP but only very rarely does it outperform Leverage SHAP.
>
> > (Yang & Salman, 2019)
>
> The only paper we could find with these two authors + year is the paper “A Fine-Grained Spectral Perspective on Neural Networks”, which we do not believe is relevant to Shapley value estimation. Perhaps you meant a different reference? Please let us know.
>
> > The technical novelty in proving the new theoretical results also appears to be limited...
>
> We do not consider the proof of our Theorem 1.1 to be a main technical contribution of the paper. It is well known that leverage score sampling provides a sample complexity bound of $O(n \log n + n/\epsilon)$ for general active linear regression problems, of which the Shapley value estimation problem can be viewed as a special case. Such results date back to work by Sarlos (FOCS, 2006) and others: a good reference is the book Sketching as a Tool for Numerical Linear Algebra (Woodruff, 2014). While our result requires some extra work (to handle paired sampling and sampling without replacement), the proof is a straightforward generalization of this prior work.  Theorem 1.1 from Shimizu et al. could likely be used as well since our paired-sampling distribution satisfies the $\ell_\infty$- independence condition in that paper. However, that result naively only applies to $k$-homogenous sampling distributions, which our distribution is not (since we do not sample a fixed number of rows from $A$). Modifying our approach to match the Shimizu result is likely possible, although ultimately it was simpler to just prove the guarantee we needed directly.
>
> Instead, we consider the technical contribution of the paper to be the realization that 1) leverage score sampling can be directly applied to the Shapley value estimation problem and 2) this can be done in a computationally efficient way given the structure of the matrix $A$. The second point is important, as applying prior work on leverage scores directly would suggest the need to compute these scores, of which there are $2^n$ for the Shapley value estimation problem.
>
> > Typos
>
> Fixed, thank you!

---

> > ### Author Response · Authors · 2024-11-19
> > **Comparison to Permutation SHAP**
> >
> > We have added a comparison to the permutation algorithm (Strumbelj & Kononenko, 2010 and Mitchell et al., 2022), which we present below. We find that Permutation SHAP sometimes outperforms Optimized Kernel SHAP but only very rarely does it outperform Leverage SHAP.
> >
> > | | IRIS | California | Diabetes | Adult | Correlated | Independent | NHANES | Communities |
> > |---------------------------|-------------|------------|-----------|------------|------------|-------------|----------------|-------------|
> > | *Optimized Kernel SHAP* | | | | | | | | |
> > | Mean | 3.28e-14 | 0.00248 | 2.33 | 1.81e-05 | 0.000739 | 0.000649 | 0.00551 | 21.8 |
> > | 1st Quartile | 2.12e-14 | 0.000279 | 0.549 | 2.16e-06 | 0.00027 | 0.000187 | 0.000707 | 5.85 |
> > | 2nd Quartile | 3.55e-14 | 0.00138 | 1.26 | 5.43e-06 | 0.000546 | 0.000385 | 0.0024 | 13.0 |
> > | 3rd Quartile | 4.22e-14 | 0.0036 | 3.03 | 1.63e-05 | 0.00101 | 0.000964 | 0.00665 | 25.1 |
> > | *Permutation SHAP* | | | | | | | | |
> > | Mean | 0.000526 | 0.000968 | 2.53 | 1.77e-05 | 0.000698 | 0.000806 | 0.00499 | 28.4 |
> > | 1st Quartile | 4.06e-14 | 0.000138 | 0.293 | 9.31e-07 | 8.79e-05 | 9.32e-05 | 0.000359 | 5.41 |
> > | 2nd Quartile | 5.57e-10 | 0.000402 | 0.968 | 3.51e-06 | 0.000428 | 0.000402 | 0.00144 | 14.7 |
> > | 3rd Quartile | 7.04e-06 | 0.00101 | 3.14 | 1.04e-05 | 0.000917 | 0.001 | 0.00438 | 36.3 |
> > | *Leverage SHAP* | | | | | | | | |
> > | Mean | 3.28e-14 | 0.000186 | 0.63 | 5.21e-06 | 0.000458 | 0.000359 | 0.00385 | 14.7 |
> > | 1st Quartile | 2.12e-14 | 1.91e-05 | 0.0631 | 6.3e-07 | 0.000139 | 9.51e-05 | 0.000333 | 3.6 |
> > | 2nd Quartile | 3.55e-14 | 8.31e-05 | 0.328 | 2.33e-06 | 0.000376 | 0.000235 | 0.00149 | 8.9 |
> > | 3rd Quartile | 4.22e-14 | 0.000231 | 0.769 | 7.09e-06 | 0.000617 | 0.000556 | 0.00401 | 15.3 |

---

> > ### Author Response · Authors · 2024-11-19
> > **Answers to Questions**
> >
> > Due to space constraints, we answer your questions here.
> >
> > > Which algorithm in the literature does the Optimized Kernel SHAP in the experiments correspond to?
> >
> > The Optimized Kernel SHAP algorithm is the Kernel SHAP implementation in the SHAP repository. The algorithm originally comes from the SHAP paper by Lundberg and Lee in 2017. It was modified to include paired sampling (a.k.a. antithetical sampling) after work by Covert and Lee in 2021. We will clarify this in the paper.
> >
> > > Do the theoretical guarantees for leverage SHAP continue to hold when optimizations like paired sampling, and sampling without replacement are not applied? Is there a difference in the guarantees with and without the optimizations?
> >
> > Yes,  the exact same theoretical guarantees for Leverage SHAP still hold even without paired sampling and sampling without replacement. However, the optimizations have a significant impact on experimental efficiency. It would be great if this difference was reflected in the theory, although we suspect that it likely comes down to constant factors in the analysis, which are difficult to get sharp (since, e.g., we rely on matrix concentration bounds without sharp constants).
> >
> > > In Table 2, Leverage SHAP w/o Bernoulli sampling seems to outperform in some cases. Is there a way to understand this? An analysis or discussion around this would be very useful to better understand the method's behavior.
> >
> > Thank you for bringing this to our attention! There was a bug in our code that was causing Leverage SHAP to take slightly fewer than $m$ samples in expectation, hurting its performance in comparison to Leverage SHAP w/o Bernoulli sampling. We will resolve this issue and rerun experiments. Doing so should slightly improve the results for Leverage SHAP.

---

> ### Author Response · Authors · 2024-11-19
> **Gamma in Experiments**
>
> We note that gamma is often quite small in experiments. Since computing $\gamma$ requires writing down the entire (exponentially) large regression problem, we can only compute $\gamma$ for datasets with small $n$. For each of these datasets, we train a model 100 times (from different random initializations) which induces a different vector $b$. Then we compute $\gamma = \| \| A \phi - b\|\|_2^2 / \|\| A \phi \|\|_2^2$. For all datasets, we found that the 75th percentile of $\gamma$ was less than 2. Please see below for the full summary statistics.
>
> | Dataset | $n$ | 1st Quartile | 2nd Quartile | 3rd Quartile |
> |--------------|-------|--------------|--------------|--------------|
> | IRIS | 4 | 0.000234 | 0.266 | 1.03 |
> | California | 8 | 0.158 | 0.298 | 0.449 |
> | Diabetes | 10 | 0.174 | 0.321 | 0.513 |
> | Adult | 12 | 0.180 | 0.395 | 0.703 |

---

> ### Comment · Reviewer_7UDD · 2024-11-25
>
> I thank the authors for clarifying my questions and appreciate the updates and responses to my review. I have updated my score based on the rebuttal.

---

### Official Review · Reviewer_LX1d · 2024-11-09

**Soundness:** 4
**Presentation:** 4
**Contribution:** 3
**Rating:** 8
**Confidence:** 4

**Summary:**

This paper provides a new algorithm for approximating Shapley values with provable guarantees. Shapley values have widespread application in ML as a way of formalizing individual feature contributions to a model's final prediction, although this paper considers the fully general setting in terms of a generic value function $v : 2^{[n]} \to \mathbb{R}$ (where $n$ is the number of features or "players").

The algorithm builds on a known way of formulating the Shapley value as the solution of a certain $2^n$-dimensional linear regression problem. The widely used "Kernel SHAP" algorithm approximates the solution by essentially subsampling the rows of the design matrix in a certain way. But a more principled approach is to subsample according to leverage scores, a well-studied concept in statistics. The catch is that naively, leverage score take time polynomial in the size of the design matrix (so $2^{O(n)}$) to compute. The key idea the authors exploit to get around this is that for the specific Shapley design matrix, the leverage scores can actually be written down in a simple closed form.

This allows them to efficiently solve the underlying regression problem and carry over known guarantees from the leverage score toolbox. Specifically, they show that the estimated Shapley values are close to the true Shapley values in a certain sense (both in terms of the optimum achieved and the values themselves). They show further refinements using paired sampling without replacement.

Finally, they show various experiments demonstrating that the resulting algorithm indeed outperforms the previous best Kernel SHAP algorithm in practice.

**Strengths:**

Shapley values are a basic and important topic in interpretable AI and beyond, finding wide application in practice. The problem of efficiently estimating them well is a very well-motivated one. This paper makes a very nice and useful contribution to this problem. The key theoretical insight of analyzing the form of the leverage scores is simple but very clever and elegant, and allows them to make use of a very well-studied toolbox in statistics (although there is still technical work to be done). It immediately feels like the "right" way to solve the problem. The resulting algorithm is theoretically sound, clean, simple, as well as effective in practice. The paper is also very clearly written, with a clear description of all the relevant context as well as clear exposition in general. I did not verify all the proofs in complete detail but they seemed correct to me.

**Weaknesses:**

I do not see any major weaknesses. I do think would be helpful for the authors to discuss the limitations of the Leverage SHAP algorithm a bit more (e.g. does it strictly dominate all prior algorithms?), and provide some context on what still remains open in this space (see below for related questions).

**Questions:**

1. It would be nice to have a fuller picture of the Shapley value problem from a broader approximation algorithms standpoint. The linear regression approach is ultimately just one approach to the problem. Do we have a sense for the best possible approximation one can hope for? Are there any computational inapproximability results?
1. Also, it's natural to ask whether additional structure makes the problem easier (e.g. for the specific setting of feature attribution). I realize this is indeed true for decision trees and such. But I am curious if the authors have an idea of a "dream result" under less restrictive but still reasonable structural assumptions, especially for feature attribution. For feature attribution for a general black box model, is the Leverage SHAP approach likely to be the best?
1. One thing that was not totally clear to me from the experiments is whether Leverage SHAP strictly dominates Kernel SHAP at every point in the running time vs accuracy graph. That is, for any fixed running time budget, is it always better to run Leverage SHAP? Theoretically, one concern could be that Leverage SHAP necessarily requires sampling $m = \Omega(n \log n)$ rows (IIUC based on Thm 1.1), whereas I believe Kernel SHAP allows you to pick $m$ arbitrarily (albeit without a guarantee). Thus perhaps for small running time budgets, maybe Kernel SHAP can sometimes be more effective than Leverage SHAP. Or perhaps Kernel SHAP is just better optimized as a practical implementation.
I think Figure 3 (sample size $m$ vs error) nearly answers this question, but my question is whether there is any subtlety in how $m$ translates to actual running time. And in general if there is any catch to the "strictly dominates Kernel SHAP" question.

Minor nit: in two places (lines 132 and 266) there is a typo: "principal" -> "principle".

---

> ### Author Response · Authors · 2024-11-19
>
> Dear Reviewer LX1d,
>
> Thank you for your time and feedback! We address your questions below.
>
> > It would be nice to have a fuller picture of the Shapley value problem from a broader approximation algorithms standpoint …
>
> Computationally, the dominant cost in estimating Shapley values for applications like feature explanation is almost always computing the value function, $v(S)$. As such, our work and much of the prior work, focuses on reducing the number of sets for which the function might need to be evaluated to estimate $\phi_i$ for all $i$. The most relevant papers that propose methods for doing so are discussed in Section 1.1. For example, Strumbelj & Kononenko 2010 and Mitchell et al. 2024 reuse samples by selecting a random permutation $S_1 \subseteq S_2 \subseteq … \subseteq S_n$ and using $v(S_i)$ to evaluate both $v(S_{i+1}) - v(S_i)$ and $v(S_{i}) - v(S_{i-1})$ when computing a standard Monte Carlo estimate of the summation form of the Shapley values (our Equation 1). In general, the regression approach taken by KernelSHAP tends to be more efficient than alternative approaches like this, which is why we focus on improving that method.
>
> > Are there any computational inapproximability results?
>
> This is an interesting question. As mentioned, since evaluating the value function, $v$, generally dominates the complexity of computing Shapley values, we believe a query complexity model would be a natural way to address this question – i.e., how many queries to to $v$ are required to estimate the Shapley values to some given level of accuracy? In this model, $\Omega(n)$ is a natural lower bound on the number of function evaluations needed to exactly compute the Shapley values, and we conjecture that this lower bound could be improved to $\Omega(n/\epsilon)$ to obtain a $(1+\epsilon)$-approximation in the sense provided in our paper.
>
> Here’s an informal argument for the $\Omega(n)$ lower bound: Consider the special linear setting where the set function is given by $v(S) = \sum_{i \in S} \alpha_i$ for some set of weights $\alpha_1, \ldots, \alpha_n$. In this setting, it can be seen that the Shapley values are exactly equal to $\alpha_1, \ldots, \alpha_n$. So, we must learn these weights exactly to learn the Shapley values. If we query $v(S)$ for $< n$ choices of $S$, we obtain a linear system with more unknowns than equations, so cannot determine the values of $\alpha_1, \ldots, \alpha_n$.
>
> As for obtaining a stronger lower bound, a natural starting point would be the $\Omega(n/\epsilon)$ lower bound proven for general active linear regression problems in the COLT 2019 paper “Active Regression via Linear-Sample Sparsification” by Chen and Price. This at least rules out a better algorithm based on linear regression – we will have to think more if it can be extended to a general sample complexity lower bound for Shapley value estimation. It’s a great question for future work!
>
> > But I am curious if the authors have an idea of a "dream result" under less restrictive but still reasonable structural assumptions…
>
> Our $O(n \log n + n /\epsilon)$ guarantee is already quite close to what we conjecture is optimal, so we think of this guarantee as almost a “dream result”. We think it would interesting to consider if other problem "structure" can lead to even better sample complexity, although for now we have focused on the general problem. Shapley values are used for a wide variety of machine learning models and Shapley value estimation algorithms like KernelSHAP are often treated as a "black-box", so as a first step, it's important to obtain results that work without additional assumptions.
>
> One additional “dream” would be to remove the $\gamma$ factor in the $\ell_2$-approximation guarantee. We believe this factor is inherent for regression-based approximation algorithms (as experimentally confirmed in Figures 5 and 6). However, we could imagine there is a non-regression-based algorithm that does not have a dependence on the quality of the optimal regression solution, as measured by $\gamma$.
>
> > One thing that was not totally clear to me from the experiments is whether Leverage SHAP strictly dominates Kernel SHAP at every point in the running time vs accuracy graph…
>
> All of the regression-based methods (including Optimized Kernel SHAP and Leverage SHAP) need at least $n$ samples so that the approximate regression problem has full rank, hence all our experiments are run with $m \geq n$. In these experiments across all eight datasets, we find that Leverage SHAP always dominates Optimized Kernel SHAP (except for cases when both methods manage to obtain the exact Shapley values up to machine precision, as in Column 1 of Table 1). You are correct that the *guarantee* on Leverage SHAP requires $m = \Omega(n \log n)$, but the method can still be run for smaller values of $n$, and we find experimentally that its performance remains superior for all $m \geq n$.
>
> Due to space constraints, we continue in a comment.

---

> > ### Author Response · Authors · 2024-11-19
> >
> > >  I do think it would be helpful for the authors to discuss the limitations of the Leverage SHAP algorithm a bit more (e.g. does it strictly dominate all prior algorithms?)...
> >
> > We find experimentally that Leverage SHAP dominates even Optimized Kernel SHAP on every dataset and setting of $m$. However, a limitation of the Leverage SHAP algorithm, and regression-based algorithms including Optimized Kernel SHAP more generally, is that the quality of the approximation depends on the optimal regression solution. This is reflected by the $\gamma$ term in the $\ell_2$-norm approximation, and all regression-based algorithms have an empirical dependence on this factor as shown in Figures 5 and 6.
> >
> > > Minor nit: in two places (lines 132 and 266) there is a typo: "principal" -> "principle".
> >
> > Fixed, thank you!

---

> > > ### Comment · Reviewer_LX1d · 2024-11-21
> > >
> > > Thank you for the responses. What I meant by [hardness of approximation](https://en.wikipedia.org/wiki/Hardness_of_approximation) is formal results establishing e.g. NP-hardness for approximating the solution up to some factor. The number of calls to $v$ seems to be only naively upper bounded by $2^n$ and lower bounded by $\Omega(n)$. This is a vast gap. What for example is the complexity of obtaining a constant factor approximation? Is it feasible in polynomial time (assuming $v$ can be evaluated in polynomial time for all $S$)? Or is it computationally hard? Hope the question makes more sense now.
> > >
> > > The other responses are satisfactory and I will keep my score.

---

> > > > ### Author Response · Authors · 2024-11-22
> > > > **Response to Hardness Clarification**
> > > >
> > > > Thank you for clarifying. To ask about computational hardness of approximation, we need to be a bit more concrete about what exactly we mean by a “constant factor approximation”. In your setting, Theorem 1.1 would already imply a polynomial time algorithm for finding approximate Shapley values $\tilde{\phi}$ that satisfy $||Z\tilde{\phi} - y|| < C ||Z \phi - y||$ for constant C. I.e., when the Shapley value problem is viewed as an optimization problem, we provide a constant factor approximation in polynomial time.
> > > >
> > > > On the other hand, suppose we wanted to, e.g., ensure that $\frac{1}{C} \phi_i \leq \tilde{\phi}_i \leq C \phi_i$ for all $i$. We claim that obtaining this goal for any constant $C$ is NP-hard in many settings. In particular, to ask about computational hardness, we also need to to be more concrete about what the input to the algorithm is: i.e., is $v$ given as a polynomial size circuit, a polynomial size formula, or as a black-box, unit cost oracle? In all of these settings the problem is NP-hard to approximate. In particular, consider the case when $v$ either evaluates to $0$ for all sets, or evaluates to $0$ for all sets except for a single set $S$. In the first case, the Shapley values will all equal $0$, whereas in the second case, the Shapley values for indices in $S$ are non-zero. As such, to obtain a multiplicative approximation, we need to determine if there is some set $S$ for which $v$ does not evaluate to 0. When $v$ is a black-box, finding such a set clearly requires $\Omega(2^n)$ time (we can only enumerate all possible inputs). However, it is still hard when $v$ is given other forms. For example, if $v$ is a given as a circuit, then determining if there is an S for which $v(S)\neq 0$ is equivalent to the NP-complete [circuit SAT problem](https://en.wikipedia.org/wiki/Circuit_satisfiability_problem).
> > > >
> > > > We will add comments on this in the next version of the paper, as others might have the same question. This argument also makes it clear why asking for a multiplicative approximation is not reasonable, and why instead we might care about an approximation in objective value. This is the case for many computational problems: for example, in the class of NP-hard optimization problems like set cover, we do not typically ask to approximate the optimal solution itself, but instead to find another solution with nearly the same objective value as the optimal solution (whether or not it is similar to that optimal solution or not).

---

> > > > > ### Comment · Reviewer_LX1d · 2024-11-25
> > > > >
> > > > > Thank you for the further clarifications. It seems like there are nontrivial subtleties in properly formulating the problem in a way that gets at what we want (IMO, that $v$ is a black box oracle and that we care most about query complexity). I definitely agree of course that in many approximation algorithm settings one wants an approximation to the objective function and not to the solution itself. It is just that Shapley values are an unusual case where arguably the "primary" definition is direct (namely Eq 1), and only as a secondary matter do we start to view it as a solution of a certain optimization problem (Eq 2). Somehow the objective function seems a bit more post-hoc / less canonical here.
> > > > >
> > > > > In any case, the takeaway for me is that perhaps a hardness of approximation view is not so straightforward. I do think some of this discussion would be suitable somewhere in the paper or an appendix.

---

### Meta-Review · Area_Chair_VyBz · 2024-12-10

**Metareview:**

This work considers the important Shapley value estimation problem. Authors show that for specific design matrices, there is a closed-form solution that can be used for leverage score sampling, which reduces the computational cost by orders of magnitude. All reviewers agree that the paper is well written and easy to follow, the proposed algorithm is interesting and significant.

**Additional Comments On Reviewer Discussion:**

Authors addressed clarity questions on their technical contribution during rebuttal. The overall rating is high, and thus there was no AC-reviewer discussion.

---

### Decision · Program_Chairs · 2025-01-22

Accept (Spotlight)